# DuRND: Rewarding from Novelty to Contribution for Reinforcement Learning via Dual Random Networks Distillation

## Abstract

Existing reward shaping techniques for sparse-reward tasks in reinforcement learning generally fall into two categories: novelty-based exploration bonuses and value-based rewards. The former encourages agents to explore less visited areas but can divert them from their main objectives, while the latter promotes stable late-stage convergence but often lacks sufficient early exploration. To combine the benefits of both, we propose Dual Random Networks Distillation (DuRND), a novel framework integrating two lightweight random network modules. These modules jointly generate two rewards: a novelty reward to drive exploration and a contribution reward to evaluate progress toward desired behaviors, achieving an efficient balance between exploration and exploitation. With low computational overhead, DuRND excels in high-dimensional environments like Atari, VizDoom, and MiniWorld, outperforming several benchmarks.

## 1 Introduction

Model-Free Reinforcement Learning (MFRL) involves an agent learning optimal policies to maximize cumulative rewards within an environment, without any prior model of its dynamics (Sutton & Barto, 2018). One pivotal challenge in MFRL is balancing exploration and exploitation, both are critical stages for effective agent learning. Sufficient exploration is vital, particularly in tasks with extremely sparse rewards where feedback is only available at the end of each episode. In such scenarios, directed exploration is necessary for agents to identify all samples that potentially yield positive effects (Ladosz et al., 2022). Conversely, in later training phases, exploitation becomes crucial to reinforce behaviors that are known to be successful in maximizing rewards, which is essential for stable convergence. Therefore, it is imperative to develop strategies that leverage information to align closely with the agent's overarching goals.

One well-studied line of work is reward shaping (RS), which designs additional rewards to supplement the sparse environmental rewards, providing fine-grained, immediate feedback (Sorg et al., 2010a;b). Introducing *exploration bonus* as auxiliary rewards stands out as a promising RS approach. By rewarding highly for novel states, it explicitly guides the agent to explore regions with insufficient experience (Baldassarre et al., 2013; Bellemare et al., 2016; Zheng et al., 2018; Devidze et al., 2022). However, since novelty does not necessarily correlate with meaningfulness or align with the agent's ultimate goals, continuously rewarding novelty may cause agents to disproportionately focus on samples from suboptimal trajectories or even dangerous regions during the stabilization stages, thereby distracting them from converging to optimal policies. The well-known "noisy TV" problem is a prime example, where agents become captivated by highly novel but irrelevant TV channels in a maze navigation task (Mavor-Parker et al., 2022). Consequently, agents need to recover from novelty rewards and shift towards exploitation gradually.

On the other hand, hidden value based RS approaches primarily develop task-related signals to reveal the extent to which states contribute to achieving higher environmental rewards and their inherent significance, e.g., the distance to the goal state, thereby enhancing exploitation (Trott et al., 2019; Memarian et al., 2021; Ma et al., 2024b;a). Compared to the exploration-centric approaches discussed earlier, these methods rely on their backbone algorithms' exploration strategies. Although

highly efficient in exploiting known experiences, they often struggle in extremely sparse-reward environments due to the lack of directional guidance toward effective exploration.

Building on the insights from both exploration bonus and hidden value RS approaches, a natural research question arises: *How can we devise a mechanism that computes both types of rewards efficiently, with minimal computational overhead and design efforts, while seamlessly evolving the reward structure from exploration to exploitation?* To this end, and inspired by Random Network Distillation (RND), which is initially developed to measure how different a state is from those previously encountered (Burda et al., 2018), we extend this concept to propose the **Du**al **R**andom **N**etworks **D**istillation (**DuRND**, pronounced "Durian") framework.

DuRND incorporates two distinct Random Network (RN) modules: a *success RN* module for states from successful trajectories and a *failure RN* module for states from failed trajectories. The sparse environmental rewards determine whether a trajectory is successful or failed. (We also extend the DuRND framework to accommodate more commonly sparse-reward scenarios, where the reward does not explicitly indicate task completion.) With the dual RN modules, we can concurrently derive two types of rewards: (a) the *novelty reward*, which evaluates how distinct a state is from all previously encountered states, and (b) the *contribution reward*, which assesses a state's historical success ratio, defined as the proportion of a state's presence in successful trajectories relative to its total occurrences. The success ratio quantifies the state's likelihood and contribution toward successful task completion or achieving high rewards, tightly aligning with the agent's objectives. Furthermore, we introduce a reward adjustment scheme that dynamically evolves from rewarding novelty to rewarding contribution as learning progresses, achieving an efficient exploration-exploitation balance. The main contributions of this paper are:

(*i*) We propose DuRND utilizing two RN modules to jointly compute two types of rewards: a novelty reward to encourage directed exploration and a contribution reward to enhance experience exploitation. By dynamically evolving the reward structure, DuRND achieves exploration-efficient and convergence-stable learning in sparse-reward tasks.

(*ii*) The rewards computation of DuRND requires lightweight computational overhead. Different from some RS methods that depend on auxiliary agents, historical states buffers, or pseudo-count estimation (Bellemare et al., 2016; Ostrovski et al., 2017; Mguni et al., 2023; Ma et al., 2024b), DuRND operates only with two RN modules, providing remarkable scalability in high-dimensional environments.

(*iii*) The effectiveness and efficiency of DuRND are validated across a variety of sparse-reward tasks with high-dimensional states, demonstrating its superior performance compared to several benchmarks.

## 2 BACKGROUND

**Reinforcement Learning (RL)** operates within the framework of **Markov Decision Processes (MDP)**, formalizing the interaction between an agent and an environment as a tuple $\langle S, A, T, R, \gamma \rangle$. $S$ and $A$ are state space and action space, respectively, $T : S \times A \times S \rightarrow [0, 1]$ is the transition function, $R : S \rightarrow \mathbb{R}$ is the reward function, and $\gamma \in [0, 1]$ is the discount factor. This paper studies stochastic policies $\pi : S \times A \rightarrow [0, 1]$ that maximize the expected discounted return $\mathbb{E}_\tau[\sum_{t=0}^{\infty} \gamma^t R(s_t)]$, where $\tau = (s_0, a_0, s_1, a_1, \ldots)$ is a trajectory of states and actions, and $s_{t+1} \sim T(\cdot|s_t, a_t)$, $a_t \sim \pi(\cdot|s_t)$. Common techniques in model-free RL encompass value-based methods, policy-based methods, and their hybrid, actor-critic methods (Sutton & Barto, 2018).

**Random Network Distillation (RND)** motivates agents to explore the less frequently visited states by incorporating novelty as an exploration bonus (Burda et al., 2018). RND introduces two neural networks: a fixed and randomly initialized *target network* $f(o) : \mathcal{O} \rightarrow \mathbb{R}^k$, and a trainable *predictor network* $\hat{f}(o; \theta) : \mathcal{O} \rightarrow \mathbb{R}^k$. Both networks map an observation $o \in \mathcal{O}$ to a $k$-dimensional feature embedding. The predictor network is trained to minimize the mean squared error (MSE) $e = \|\hat{f}(o; \theta) - f(o)\|^2$ through gradient descent. This MSE for a specific observation $o$ is also used to quantify its novelty, as higher errors are expected for states that are dissimilar to those the predictor has been trained on previously, thereby the exploration bonus is defined as $r^{\text{rnd}} = e$. As the predictor is trained with samples collected by the agent, it gradually develops a "memory" of the states it has seen. RND has proven effective in assessing novelty to encourage exploration.

## 3 RELATED WORK

**Exploration Bonuses** as shaping rewards have been widely used to guide the exploration directions. The most intuitive method is the count-based approach, where the novelty of each state is assessed by its visitation frequency (Strehl & Littman, 2008). To adapt state counting to continuous or unlimited state spaces, pseudo-counts were introduced (Bellemare et al., 2016), with several works studied how to estimate the pseudo-counts (Fox et al., 2018; Badia et al., 2020; Devidze et al., 2022). Specifically, Bellemare et al. (2016) derived from the Context Tree Switching model, Fu et al. (2017) used exemplar models for implicit density estimation, Tang et al. (2017) discretized continuous states using hash functions, and Machado et al. (2020) incorporated the successor representation. Although tractable, these methods often require extensive storage resources or inference time. Following the pseudo-count concept, neural network-based methods have been developed. Ostrovski et al. (2017) used PixelCNN (Van den Oord et al., 2016) for density estimation; Martin et al. (2017) used the feature representation from value approximation networks; Lobel et al. (2023) derived the pseudo-counts by averaging samples from the Rademacher distribution; and Burda et al. (2018) introduced Random Network Distillation to assess state novelty, while Yang et al. (2024) further improved the precision of bonus allocation. Our work extends the RND approach to efficiently count state visitations in high-dimensional spaces.

**Hidden Values** as shaping rewards effectively guide the optimization direction of agents to accelerate the convergence. One common approach is to extract reward models from expert demonstrations (Inverse RL) (Arora & Doshi, 2021; Cheng et al., 2021) or human feedback (RLHF) (Christiano et al., 2017), which have been popularly applied in robotic control (Ellis et al., 2021; Schultheis et al., 2021; Bıyık et al., 2022) and large language models (LLMs) (Sumers et al., 2021; Ghosal et al., 2023; Wu et al., 2023; Hwang et al., 2023; Dai et al., 2024). However, these methods require considerable human-generated data, which is often challenging to obtain, especially in highly specialized or advanced domains. Another line of research has emerged to derive beneficial information directly from the agent's own learning experiences (Zheng et al., 2018; Hu et al., 2020; Park et al., 2023; Gupta et al., 2023). Representatively, Trott et al. (2019) used the state-goal distance as heuristics, Memarian et al. (2021) ranked different trajectories via a trained classifier indicated by the preferences, Ma et al. (2024b) introduced an assistant reward agent to collaboratively generate rewards guiding the policy agent, Ma et al. (2024a) derived the success ratio based on Thompson sampling framework to evaluate a state's contribution to task completion. However, although these methods effectively accelerate agent convergence, their reliance on the underlying algorithm's exploration strategies may lead to suboptimal policies due to insufficient sample diversity. Our work seeks to combine the shaping rewards of hidden value with exploration bonuses, aiming to achieve efficient exploration and fast convergence.

Other reward shaping methods have been explored, leveraging diverse strategies. Potential-based algorithms defined rewards as the temporal difference of a potential function, ensuring that the optimal policy remains consistent with the original MDP (Asmuth et al., 2008; Devlin & Kudenko, 2012). Information gain based approaches used the prediction errors in dynamics to model how surprising the states are to motivate exploration (Houthooft et al., 2016; Pathak et al., 2017; Hong et al., 2018; Burda et al., 2019; Sun et al., 2022). However, both branches require an environmental transition model, which makes them challenging in adapting to large-scale scenarios with complex dynamics. Additionally, some studies incorporated concepts of uncertainty or diversity (Eysenbach et al., 2019; Pathak et al., 2019; Raileanu & Rocktäschel, 2020), or involved multiple agents or hierarchical structures to shape rewards (Stadie et al., 2020; Yi et al., 2022; Mguni et al., 2023).

## 4 METHODOLOGY

### 4.1 OVERVIEW OF THE DuRND FRAMEWORK

In our DuRND framework, the shaping reward is defined by integrating two auxiliary rewards:

$$R^{\text{DuRND}}(s) := R^{\text{env}}(s) + \lambda R^{\text{nov}}(s) + \omega R^{\text{con}}(s), \tag{1}$$

where $\lambda$ and $\omega$ are parameters that control the relative scales of the rewards. Here, $R^{\text{env}}(s)$ is the environment reward, $R^{\text{nov}}(s)$ is the *novelty reward*, serving as the exploration bonus, and $R^{\text{con}}(s)$ is the *contribution reward*, which assesses the states' hidden value in achieving overall performance.

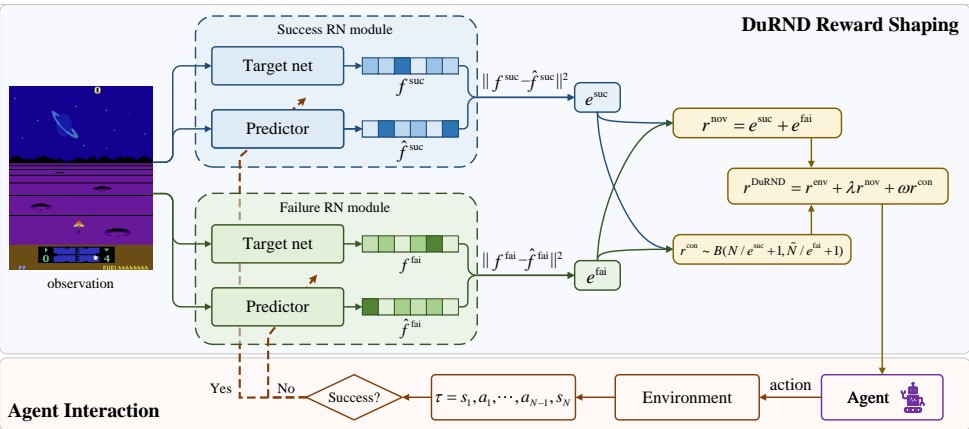

Figure 1: An overview of the Dual Random Networks Distillation (DuRND) framework. The observation is processed through both Success and Failure RN modules to derive errors that reflect its novelty in successful and failed scenarios, respectively. The two errors jointly form the DuRND shaping rewards used to train the agent. At the end of each trajectory, the corresponding RN module is updated based on the trajectory's outcome, as indicated by the sparse environmental reward.

Both $R^{\mathrm{nov}}(s)$ and $R^{\mathrm{con}}(s)$ are jointly computed by two distinct Random Network (RN) modules, referred to as the *success RN* and the *failure RN*. They are updated based on successful and failed trajectories, respectively, throughout the training process. A high-level overview of the DuRND framework is illustrated in Figure 1.

## 4.2 REWARD SHAPING VIA DUAL RANDOM NETWORKS

### 4.2.1 DUAL RANDOM NETWORK MODULES

We introduce two distinct RN modules: the success RN module $\mathcal{R}_S$ and the failure RN module $\mathcal{R}_F$. Each module consists of two separate networks: a fixed and randomly initialized target network $f_X(o) : \mathcal{O} \rightarrow \mathbb{R}^k$, and a differently initialized predictor $\hat{f}_X(o; \theta_X) : \mathcal{O} \rightarrow \mathbb{R}^k$, parameterized by $\theta_X$, where $X \in \{S, F\}$. It is worth noting that to prevent estimation bias from differences between the two modules, both the architecture and weights of the target networks in $\mathcal{R}_S$ and $\mathcal{R}_F$ are identical. Similarly, the predictors in both modules are also initialized identically.

At the end of each episode, samples from the entire trajectory are used to update the corresponding RN module, identified as successful or failed based on environmental rewards. The criteria for trajectory classification are further detailed in Section 4.2.3. Specifically, for a given trajectory of states $\tau_X = \{s_1, s_2, \ldots, s_T\}$, the predictor is updated to minimize the MSE loss:

$$e_X(s_t; \theta_X) = \left\| f_X(s_t) - \hat{f}_X(s_t; \theta_X) \right\|^2, \qquad \forall s_t \in \tau_X, \quad X \in \{S, F\}. \tag{2}$$

By updating the predictors with the states observed by the agent, we harness the epistemic uncertainty inherent in deep neural network training, where error progressively decreases as the volume of training data increases (Burda et al., 2018). Consequently, this error, $e_X$, itself effectively functions as a density estimation for the states previously encountered by the agent, with larger errors indicating less frequently visited states. Moving forward, we introduce how the two RN modules collaboratively compute the two types of rewards.

### 4.2.2 NOVELTY AND CONTRIBUTION REWARDS

**Novelty Reward.** Since all historical states are delivered to update either $\mathcal{R}_S$ or $\mathcal{R}_F$, the novelty of a state regarding all previously encountered samples is naturally assessed by combining the prediction errors from both modules, thus the novelty reward is defined as:

$$R^{\mathrm{nov}}(s) = e_S(s) + e_F(s), \tag{3}$$

where $e_S$ and $e_F$ are the prediction errors from $\mathcal{R}_S$ and $\mathcal{R}_F$, respectively, calculated by Equation 2.

**Contribution Reward.** To evaluate the hidden value of a state, we consider its *success ratio*, which is defined as the proportion of times a state appears in successful trajectories relative to its total historical occurrences. In sparse-reward environments, a higher success ratio signifies a state's greater likelihood of contributing to successful task completion, aligning closely with the agent's overall objective. Given the prediction errors $e_S(s)$ and $e_F(s)$, which are related to the respective state's *infrequency*, the historical success ratio SR is estimated as:

$$\text{SR}(s) = \frac{1}{e_S(s)} \Big/ \Big( \frac{1}{e_S(s)} + \frac{1}{e_F(s)} \Big) = \frac{e_F(s)}{e_S(s) + e_F(s)}. \tag{4}$$

However, directly using a static success ratio may lead to local optima due to premature overconfidence, as noted by Ma et al. (2024a). To address this, a Beta distribution is constructed from which we sample the contribution reward. The Beta distribution for a specific state involves two parameters, $\tilde{N}_S(s)$ and $\tilde{N}_F(s)$, which are positively correlated with the actual pseudo-counts for state $s$ encountered in successful and failed trajectories, respectively, based on the RN errors, can be estimated as:

$$\tilde{N}_X(s) = \frac{N(t)}{e_X(s)}, \qquad X \in \{S, F\}, \tag{5}$$

where $N(t)$ is the total number of states observed by the agent up to time $t$. Then the contribution reward is derived as:

$$R^{\text{con}}(s) = \hat{r}, \hat{r} \sim \text{Beta}(r; \tilde{N}_S(s) + 1, \tilde{N}_F(s) + 1) = \frac{r^{\tilde{N}_S(s)}(1 - r)^{\tilde{N}_F(s)}}{B\big(\tilde{N}_S(s) + 1, \tilde{N}_F(s) + 1\big)}, \tag{6}$$

where $r$ denotes the random variable and $B(\cdot, \cdot)$ is the normalization factor.

The theoretical foundation for using Beta distribution is supported by Thompson Sampling framework (Thompson, 1933). A key intrinsic property of the Beta distribution is that as the sample size increases, i.e., $\tilde{N}_S(s)$ and $\tilde{N}_F(s)$ grow, it gradually converges to the true success ratio, demonstrating an adaptive convergence in response to increasing confidence level. Notably, due to the non-linear nature of neural networks, the pseudo-counts derived from RN modules are not necessarily linear to the actual counts. However, this discrepancy does not compromise our approach as we primarily require a relative measure of success and failure counts, not precise values.

Finally, to effectively balance exploration and exploitation, we dynamically adjust the weights of the novelty and contribution rewards, $\lambda$ and $\omega$, in Equation 1. We set $\lambda$ to decrease linearly from 1 to 0, and $\omega$ to increase from 0 to 1 throughout the training process, which has been validated to be effective in practice. This ensures that the agent initially focuses on exploration and gradually shifts to exploitation by strategically scheduling the two rewards.

### 4.2.3 SUCCESSFUL AND FAILED TRAJECTORIES

In our DuRND framework, trajectories are classified as successful or failed based on environmental rewards. For scenarios with task-completion indication rewards, which are typically issued at the end of a trajectory, success is straightforwardly inferred from the final reward. For example, in a maze navigation task, a reward is given only upon reaching the destination.

To extend DuRND to environments where rewards do not directly indicate overall trajectory success, yet are still sparse and assigned to key milestones or sub-goals, we introduce a sub-trajectory approach. This strategy is based on a fundamental assumption that a trajectory can be divided into multiple sub-trajectories, each independently labeled as successful or not. The cumulative reward obtained by the entire trajectory is then considered a collective contribution of all these sub-trajectories. We define a hyperparameter $T_{\max}$, representing the maximum length for a sub-trajectory. If no reward is received within $T_{\max}$ consecutive steps, the sub-trajectory is considered failed; conversely, receiving any reward within $T_{\max}$ marks the preceding sequence as a successful sub-trajectory. This approach enables DuRND to flexibly adapt to more general sparse-reward structures.

### 4.3 DURND ENHANCED RL ALGORITHM

We integrate the DuRND framework into Proximal Policy Optimization (PPO), a well-known, advanced on-policy RL algorithm (Schulman et al., 2017). PPO consists of two modules: a policy

to select actions given states (actor) and a value function to evaluate the policy's behavior (critic). The enhancement is to use the DuRND-defined reward structure in Equation 1 to shape the sparse environmental rewards. Let $\pi_\theta$ be the parameterized policy network and $V_\phi$ be the parameterized Value network. We define the enhanced *advantage* given the DuRND reward as:

$$\hat{A}_t = \sum_{l=0}^{T-t-l} \gamma^l \delta_{t+l}, \qquad \delta_t = \left(r_t^{\text{env}} + \lambda r_t^{\text{nov}} + \omega r_t^{\text{con}}\right) + \gamma V_{\phi_{\text{old}}}(s_{t+1}) - V_{\phi_{\text{old}}}(s_t). \tag{7}$$

Then the enhanced loss function for policy $\pi_\theta$ is defined as:

$$\hat{L}(\theta) = \mathbb{E}\left[ \min\left( r_t(\theta)\hat{A}_t, \text{clip}\left(r_t(\theta),\, 1-\epsilon,\, 1+\epsilon\right)\hat{A}_t \right) \right], \tag{8}$$

where $r_t(\theta) = \dfrac{\pi_\theta(a_t|s_t)}{\pi_{\theta_{\text{old}}}(a_t|s_t)}$ is the probability ratio, and $\epsilon$ is the clipping parameter. The enhanced loss function for the value function is defined as:

$$\hat{L}(\phi) = \mathbb{E}\left[ \left( V_\phi(s_t) - \left(\hat{A}_t + V_{\phi_{\text{old}}}(s_t)\right) \right)^2 \right]. \tag{9}$$

By leveraging real-time computed novelty and contribution rewards, alongside their linearly updated weights, the augmented DuRND rewards effectively broaden the exploration horizon in early training and progressively evolve to density, meaningful rewards later, improving convergence. We implement the DuRND framework within the PPO algorithm, primarily following the vanilla RND model (Burda et al., 2018). The trajectory-based optimization nature of PPO also fits well with the DuRND's updates. Besides, DuRND can be easily adapted to more model-free RL algorithms, such as SAC (Haarnoja et al., 2018), TD3 (Fujimoto et al., 2018), and others. We summarize the DuRND-enhanced PPO algorithm in Algorithm 1.

---

**Algorithm 1** Dual Random Networks Distillation enhanced Proximal Policy Optimization

---

**Require:** Environment $\mathcal{E}$, parameterized $\pi_\theta$ and $V_\phi$
**Require:** Random Network modules $\mathcal{R}_S$ and $\mathcal{R}_F$
1: **for** iteration $= 1, 2, \dots$ **do**
2:    **for** each epoch and $\mathcal{T} = \emptyset$ **do**
3:       $(s_t, a_t, r_t^{\text{env}}, s_{t+1}) \leftarrow \text{Interact}(\pi_{\theta_{\text{old}}}, \mathcal{E})$    ▷ collect transitions by executing current policy
4:       $e_S(s_t) \sim \mathcal{R}_S, e_F(s_t) \sim \mathcal{R}_F$    ▷ compute prediction errors from two RN modules
5:       $r_t^{\text{nov}} = e_S(s_t) + e_F(s_t)$    ▷ compute novelty reward
6:       $r_t^{\text{con}} \sim \text{Beta}(r; N(t)/e_S(s_t) + 1, N(t)/e_F(s_t) + 1)$    ▷ sample contribution reward
7:       $\mathcal{T} \leftarrow \mathcal{T} \cup \{(s_t, a_t, r_t^{\text{new}}, r_t^{\text{nov}}, r_t^{\text{con}}, s_{t+1})\}$    ▷ store augmented transitions
8:    **end for**
9:    **if** trajectory is successful: $\mathcal{R}_S \leftarrow \text{Update}(\mathcal{R}_S, \mathcal{T})$    ▷ update success RN module
10:    **else:** $\mathcal{R}_F \leftarrow \text{Update}(\mathcal{R}_F, \mathcal{T})$    ▷ otherwise, update failure RN module
11:    $\theta \leftarrow \theta - \alpha_\theta \nabla_\theta \hat{L}(\theta)$    ▷ optimize $\pi_\theta$ by Equation 8
12:    $\phi \leftarrow \phi - \alpha_\phi \nabla_\phi \hat{L}(\phi)$    ▷ optimize $V_\phi$ by Equation 9
13: **end for**

---

## 5 EXPERIMENTS

Experiments are designed to evaluate the DuRND framework across various environments with sparse rewards. We select ten challenge tasks from three domains: *Atari games*, classic 2D games from the arcade learning environment (ALE) platform (Bellemare et al., 2013), *VizDoom*, a 3D first-person shooting game environment (Kempka et al., 2016; Tomilin et al., 2022), and *Mini-World*, a simulated 3D interior maze environment (Chevalier-Boisvert et al., 2023). Specifically, the two MiniWorld environments provide task-completion indication rewards only at the end of each episode, while other environments assign rewards for achieving specific milestones, but the overall distribution of rewards remains highly sparse. To ensure consistency in reward scaling across all environments, we standardize rewards: 1 for task completion or milestone achievement and 0 otherwise. Illustrations of all tasks can be found in Figure 2, with detailed descriptions of these tasks and the environmental reward structures provided in Appendix A.1.

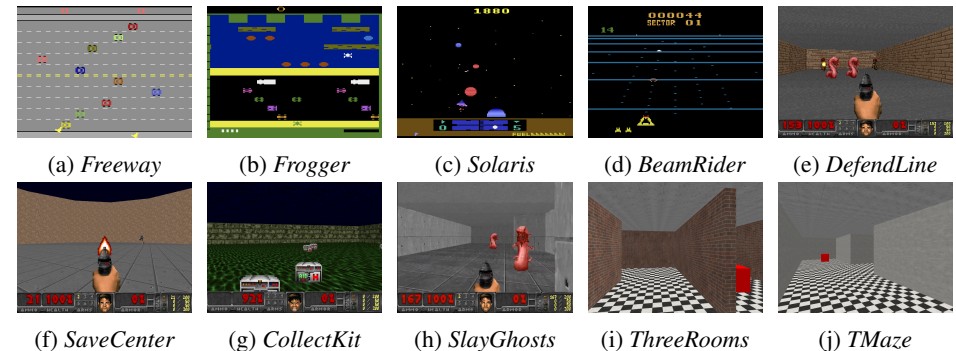

Figure 2: Ten sparse-reward tasks evaluated in our experiments.

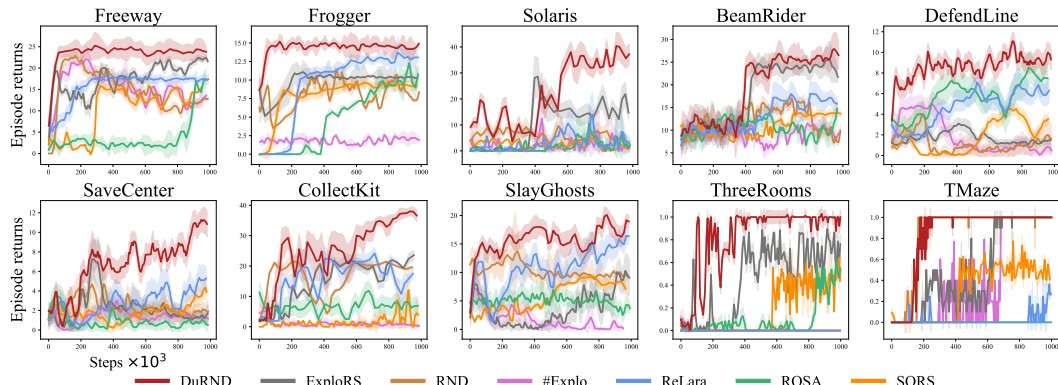

Figure 3: The learning performance of DuRND compared with baselines.

## 5.1 COMPARISON TO BASELINES

We compare DuRND with six widely-recognized reward shaping baselines, covering the two main categories we have discussed. For approaches that incorporate exploration bonuses as auxiliary rewards, we include *ExploRS* (Devidze et al., 2022), *RND* (Burda et al., 2018), and *#Explo* (Tang et al., 2017); For approaches that extract hidden value-based rewards, we include *ReLara* (Ma et al., 2024b), *ROSA* (Mguni et al., 2023), and *SORS* (Memarian et al., 2021). All baselines are implemented based on the *RLeXplore* library (Yuan et al., 2024) or the codes attached in the respective papers. Further experimental details, like hyperparameters, neural network architectures, and hardware configurations, are provided in Appendix A.2.

The learning results averaged over ten runs with different random seeds, are illustrated in Figure 3, with the quantified data presented in Table 1. The DuRND framework demonstrates distinct advantages mainly from three aspects: efficient and directed exploration, rapid and stable convergence, and considerably low training resource demands.

**Exploration.** DuRND inherits its exploratory capability from the RND's intrinsic exploration bonus. Rewarding novelty allows the agent to assign higher rewards to less frequently visited states, thus encouraging more targeted exploration. For the baselines, ReLara relies on random reward injections and random action space sampling that mainly introduce noise to amplify uncertainty; ROSA and SORS depend on the agent's underlying exploration strategies. All these three baselines lack explicit guidance on which areas to explore. Consequently, DuRND is observed to collect trajectories with higher episodic returns earlier in training due to the novelty reward, enhancing sample efficiency. Furthermore, while ReLara, ROSA, and SORS can also converge to optimal policies in many settings, they sometimes remain trapped in local optima. For instance, in the *SaveCenter* tasks, DuRND achieves higher returns by continuously defeating 12 enemies in one episode, while the baselines only defeat about 6 within the same training periods.

Table 1: Comparison of DuRND with baseline models: the average episodic returns with standard errors (↑ higher is better).

| Environments | DuRND | ExploRS | RND | #Explo | ReLara | ROSA | SORS |
|---|---|---|---|---|---|---|---|
| *Freeway* | **23.22 ± 0.01** | 17.46 ± 0.00 | 14.77 ± 0.01 | 15.16 ± 0.01 | 15.47 ± 0.00 | 3.68 ± 0.00 | 7.30 ± 0.01 |
| *Frogger* | **14.36 ± 0.00** | 10.19 ± 0.00 | 8.59 ± 0.00 | 1.81 ± 0.00 | 9.30 ± 0.01 | 3.45 ± 0.00 | 7.79 ± 0.00 |
| *Solaris* | **18.91 ± 0.02** | 9.82 ± 0.01 | 6.07 ± 0.00 | 2.06 ± 0.00 | 2.96 ± 0.00 | 1.87 ± 0.00 | 2.50 ± 0.00 |
| *BeamRider* | **18.05 ± 0.01** | 16.19 ± 0.01 | 11.96 ± 0.00 | 9.03 ± 0.00 | 11.84 ± 0.00 | 10.57 ± 0.00 | 10.56 ± 0.00 |
| *DefendLine* | **8.52 ± 0.00** | 1.63 ± 0.00 | 1.11 ± 0.00 | 1.62 ± 0.00 | 4.27 ± 0.00 | 5.33 ± 0.00 | 1.28 ± 0.01 |
| *SaveCenter* | **6.33 ± 0.00** | 2.03 ± 0.00 | 2.37 ± 0.00 | 1.30 ± 0.00 | 2.64 ± 0.00 | 0.83 ± 0.00 | 1.78 ± 0.01 |
| *CollectKit* | **20.87 ± 0.01** | 11.97 ± 0.01 | 14.59 ± 0.01 | 0.90 ± 0.00 | 12.43 ± 0.01 | 6.80 ± 0.00 | 1.60 ± 0.00 |
| *SlayGhosts* | **15.60 ± 0.00** | 2.82 ± 0.00 | 10.18 ± 0.00 | 1.27 ± 0.00 | 10.61 ± 0.00 | 5.01 ± 0.00 | 5.07 ± 0.01 |
| *ThreeRooms* | **0.86 ± 0.00** | 0.48 ± 0.00 | 0.00 ± 0.00 | 0.00 ± 0.00 | 0.00 ± 0.00 | 0.12 ± 0.00 | 0.18 ± 0.00 |
| *TMaze* | 0.96 ± 0.00 | 0.80 ± 0.00 | **0.97 ± 0.00** | 0.39 ± 0.00 | 0.02 ± 0.00 | 0.00 ± 0.00 | 0.30 ± 0.00 |

**Exploitation.** DuRND progressively shifts from rewarding novelty to rewarding contribution, enhancing the agent's focus on states that are more likely to result in successful task completion, thereby continuously reinforcing beneficial behaviors. However, for the baselines that only incorporate exploration bonuses, such as ExploRS, RND, and #Explo, agents struggle to derive effective guidance from novelty rewards as training progresses to later stages. More critically, the agents' overemphasis on novel yet low-value states hinders the recovery from the shaping rewards, leading to policies that diverge from the original task objectives. Observations in tasks like *Freeway*, *DefendLine*, and *SlayGhosts* reveal that while these baselines may initially achieve high environmental returns, their performance declines in later stages, deviating from the optimal policies. Conversely, DuRND maintains a steady convergence towards the optimal policy, demonstrating its effectiveness in balancing exploration and exploitation.

**Memory Efficiency.** DuRND is space-efficient as it only introduces two lightweight RN modules to compute both types of rewards. In comparison, ReLara and ROSA both demand additional agents, which are generally more complex and computationally expensive. ExploRS and #Explo both involve pseudo-counts but are not RND-based, relying instead on density models that require substantial extra space for storing historical states (at least partially). To empirically validate DuRND's memory efficiency, we report the maximum memory consumption in Table 2. To provide a more intuitive comparison, we normalize the data relative to our DuRND. To keep the comparison fair between off-policy and on-policy methods, we exclude the memory consumption of replay buffers.

Table 2: The maximum memory consumption during training across three domains, normalized relative to DuRND to report intuitively (↓ lower is better).

| Domains | DuRND | ExploRS | RND | #Explo | ReLara | ROSA | SORS |
|---|---|---|---|---|---|---|---|
| *Atari games* | 1 | 10.94 | 0.91 | 0.84 | 3.67 | 3.84 | 1.1 |
| *VizDoom* | 1 | 11.94 | 0.93 | 0.84 | 3.97 | 4.24 | 1.06 |
| *MiniWorld* | 1 | 11.41 | 0.90 | 0.83 | 3.58 | 3.91 | 1.12 |

## 5.2 EXPLORATION-EXPLOITATION TRADE-OFF

In this section, we further study the exploration-exploitation trade-off in DuRND by demonstrating the differences in state visitation distributions under different reward shaping methods and exploration strategies. For an intuitive illustration, we consider a toy task in a one-dimensional chain of length 31, with states as $s_0, s_1, \cdots, s_{30}$ from left to right. The agent starts at the midpoint, $s_{15}$, at the beginning of each episode. There are 15 states on either side of the starting point, but only the far-right state, $s_{30}$, is the successful terminal state with $R^{env}(s_{30}) = 1$, while all other states are rewarded as 0. Each episode is limited to a maximum of 20 steps. The agent can take three actions: moving to the left, moving to the right, and staying in the current state.

We compare the complete DuRND with two variants: (1) DuRND with only the novelty reward $\lambda R^{nov}$, and (2) DuRND with only the contribution reward $\omega R^{con}$; as well as three reward shaping or exploration approaches: (3) *vanilla RND*, that only rewards novelty; (4) *SORS*, that defines shaping rewards by ranking trajectories with environmental feedback; and (5) *ϵ-greedy*, the popular strategy that selects a random action with probability $\epsilon$ and the greedy action with probability $1 - \epsilon$. For each

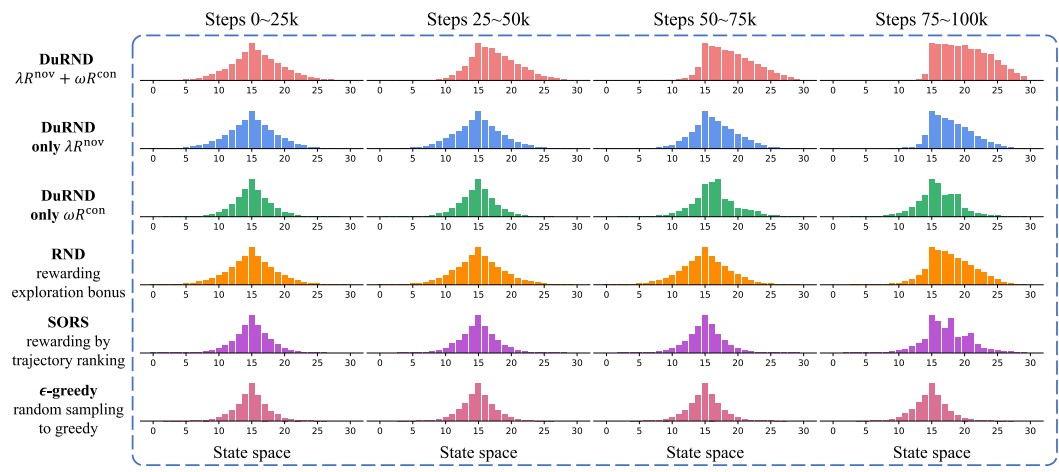

Figure 4: The state visiting distributions of different methods for each 25k steps in the toy task.

method, we track the state visitation over a total of 100k steps in the toy task, presenting the results for every 25k steps in Figure 4.

From the presented results, we observe that DuRND demonstrates an efficient trade-off between exploration and exploitation. In the early stage (around 0 to 50k steps), DuRND shows a more balanced state visitation across the entire chain, while in the later stage (around 50k to 100k steps), the agent increasingly focuses on the right side of the starting point, as only these states yield task completion. In comparison, RND maintains a broader exploratory behavior but is less effective at the exploitation stage, still visiting states on the left side even in the last 25k steps. DuRND with only $\lambda R^{\text{nov}}$ performs better than RND because of the novelty reward scheduling; however it performs worse than the complete DuRND, indicating the effectiveness of the $\omega R^{\text{con}}$ term. SORS and DuRND with only $\omega R^{\text{con}}$ converge slower than complete DuRND, and their exploration ranges are more limited. For $\epsilon$-greedy, which lacks a clear exploratory direction, the initial exploration is more concentrated, consequently, it fails to reach the terminal state within the 100k steps.

### 5.3 Novelty and Contribution Rewards

#### 5.3.1 Analysis of the Learned Rewards

We discuss how the novelty and contribution rewards evolve during training. Figure 5 shows the normalized rewards received by the agent throughout learning. Over time, the novelty reward decreases while the contribution reward increases, both nonlinearly. The decline in the novelty reward indicates the diminishing differentiation among states after extensive exploration, i.e., states become uniformly non-novel, thus, the information provided by novelty rewards loses significance in later training, highlighting again the limitation of relying only on novelty may hinder convergence. The contribution reward increases and eventually stabilizes at a high level, dominating the shaping rewards. This is attributed to the continuous reinforcement of successful trajectories, which directs the agent's focus towards states conducive to success, thereby causing the contribution rewards to converge to a stable level. In summary, the transition from exploration-driven to task-oriented rewards is a critical factor underpinning DuRND's superior performance.

#### 5.3.2 Ablation Study: Effects of Two Types of Rewards

To further understand the effects of two types of rewards, we conduct an ablation study to compare the complete DuRND framework with two variants: (1) DuRND with only the novelty reward (*only $R^{\text{nov}}$*), and (2) DuRND with only the contribution reward (*only $R^{\text{con}}$*). The learning curves are shown in Figure 6, with the quantitative results provided in Appendix A.3.

The experimental results show that both rewards are essential for DuRND's performance. When relying only on the novelty reward, agents struggle to recover the environmental rewards, leading to

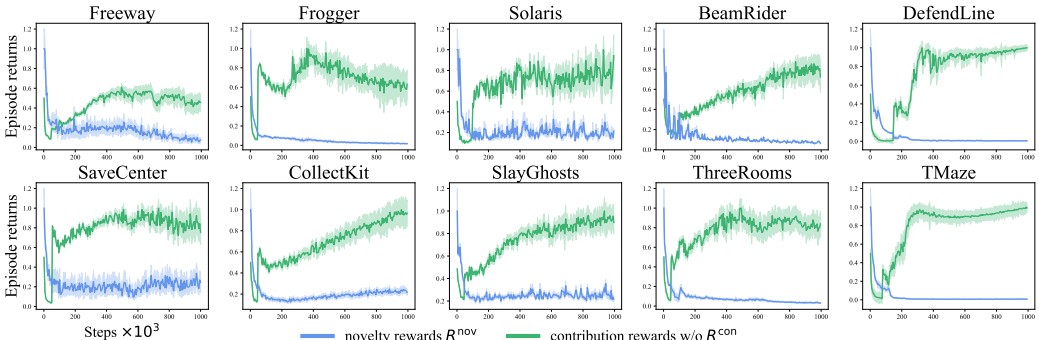

Figure 5: The novelty and contribution rewards learned in DuRND framework.

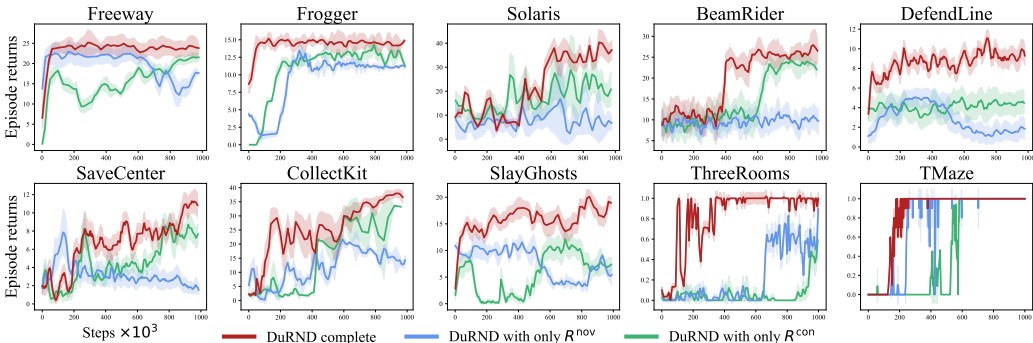

Figure 6: Ablation study: the learning performance of DuRND with a single type of reward.

unstable convergence and deviations from the task's original objectives. But this variant outperforms the vanilla RND, as the decreasing on the novelty reward over time alleviates the agent's distraction. In contrast, using only the contribution reward hinders efficient exploration, delaying favorable outcomes and potentially trapping the agent in local optima.

## 6 CONCLUSION AND DISCUSSION

**Conclusion.** This paper introduced the DuRND framework, designed to separately estimate the visitation frequencies of states from both successful and failed (sub-)trajectories. The dual RN modules compute two types of rewards, guiding the agent from directed exploration to stable convergence. Experimentally, we demonstrate that compared to the novelty-based RS approaches, DuRND avoids the pitfalls of continuous novelty-driven exploration, instead shifting to provide more meaningful rewards for desired behaviors; while compared to the hidden value based RS approaches, DuRND effectively broadens the exploration scope and collects more diverse information. In summary, DuRND combines the advantages of both approaches, achieving an efficient tradeoff between exploration and exploitation. Moreover, DuRND operates with low computational overhead in high-dimensional environments, making it a scalable solution for a wide range of RL tasks.

**Limitations.** We find that in non-task-completion-indication reward scenarios, DuRND remains sensitive to the maximum sub-trajectory length $T_{\max}$, as it affects the accuracy of classifying states as successful or failed. This hyperparameter also varies across environments, depending on the degree of reward sparsity. Thus, determining the appropriate $T_{\max}$ requires some environment-specific prior knowledge. To adapt to diverse settings, a dynamic $T_{\max}$ that adjusts according to the environment's average reward cycle could be considered. Additionally, while linearly adjusting the weights of the two rewards has been empirically effective, this approach may not be optimal. Identifying the right moment to shift from rewarding novelty to rewarding contribution may need better metrics to gauge whether exploration has been sufficient. This presents a valuable direction for future research.

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

# A APPENDIX

## A.1 ENVIRONMENTS CONFIGURATION

All tasks in our experiments provide sparse rewards. The objective descriptions and the criteria for assigning sparse environmental rewards are detailed in Table 3. Apart from tasks *ThreeRooms* and *TMaze*, which offer episodic rewards, other tasks provide intermediate rewards upon the completion of some specific milestones. All other states yield zero reward.

Table 3: Objective descriptions and environmental rewards assignments for the ten tasks.

| Environments | Sparse Rewards Assignment |
|---|---|
| *Freeway* | Guide the chicken across multiple lanes of heavy traffic.
1. +1 reward for the chicken goes across the screen.
2. Episode ends if all 3 chickens are hit by cars or maximum steps 2000 are reached. |
| *Frogger* | Guide the frog home across a highway and river while avoiding cars and predators.
1. +2 rewards for reaching home.
2. +1 reward for eating a fly.
3. Episode ends when all 5 frogs are lost or maximum steps 2000 are reached. |
| *Solaris* | Control a spaceship to blast enemies and explore new galaxies.
1. +1 reward for destroying a target.
2. +1 reward for entering a new galaxy.
3. Episode ends when all ships are destroyed or maximum steps 2000 are reached. |
| *BeamRider* | Control a spaceship to destroy enemies while avoiding obstacles.
1. +1 reward for each enemy ship destroyed.
2. Episode ends if all ships are lost or maximum steps 2000 are reached. |
| *DefendLine* | Defend the line by neutralizing incoming enemies.
1. +1 reward for each enemy killed.
2. Episode ends if the player is defeated or the maximum steps 1000 are reached. |
| *SaveCenter* | Protect the center by eliminating enemies.
1. +1 reward for each enemy killed.
2. Episode ends if the player is defeated or the maximum steps 1000 are reached. |
| *CollectKit* | Collect health kits in a room full of poison.
1. +1 reward for collecting one kit.
2. Episode ends if the player is killed by the poison or the maximum steps 1000 are reached. |
| *SlayGhosts* | Eliminate ghosts or monsters in a designated environment.
1. +1 reward for each ghost killed.
2. Episode ends if the player is killed or the maximum steps 1000 are reached. |
| *ThreeRooms* | Navigate through three connected rooms to reach a red cube.
1. +1 reward for reaching the red cube.
2. −0.1 penalty for each time step taken.
3. Episode ends when the cube is reached or the maximum steps 500 are reached. |
| *TMaze* | Navigate a T-shaped maze to reach the red cube.
1. +1 point for reaching the red cube.
2. −0.1 penalty for each time step taken.
3. Episode ends when the cube is reached or the maximum steps 500 are reached. |

## A.2 EXPERIMENTS IMPLEMENTATION DETAILS

### A.2.1 IMPLEMENTATION DETAILS

In this section, we discuss some details of the implementation of our DuRND framework.

**Observation Normalization.** Observation normalization is a common practice in deep reinforcement learning, which helps stabilize the learning process. We normalize the observations by subtracting the running mean and dividing by the running standard deviation, following the implementation introduced in Burda et al. (2018).

**Random Networks Error Normalization.** For different tasks and different initializations of the random network modules, the scale of the MSE errors, $e_S$ and $e_F$, can vary significantly. To make it easy to formalize the hyperparameter $\lambda$ across different tasks, we normalize the MSE errors by dividing them by the *initial error*, which is the average of the MSE errors from the first mini-batch at the beginning of the training process. This is built on the assumption that the errors are gradually decreasing, so the initial error is a good approximation of the scale of the errors.

**State Number Function.** In implementation, the state number estimation function $N(t)$ in Equation 5 is not directly assigned as the corresponding time step $t$. Instead, we use a factor $\phi$ to scale the state number function, which is defined as $N(t) = \phi t$, where $\phi = 0.01$ in our experiments. This is mainly because that directly using the time step $t$ results in overly large estimated pseudo-counts, which may lead to premature confidence in the Beta distributions, thus leading to suboptima.

### A.2.2 HYPERPARAMETERS

DuRND is relatively robust to hyperparameters, we report the hyperparameters used in our experiments in Table 4.

Table 4: The hyperparameters of DuRND in our experiments.

| Hyperparameters | Values |
|---|---|
| discount factor $\gamma$ | 0.99 |
| generalized advantage estimate | 0.95 |
| number of mini-batches | 32 |
| learning rate | $3 \times 10^{-4}$ |
| maximum gradient normalization | 0.5 |
| random networks learning rate | $10^{-6}$ |
| PPO clip coefficient | 0.2 |
| PPO entropy coefficient | 0.0 |
| PPO value loss coefficient | 0.5 |
| Total training steps | $10^6$ |

### A.2.3 NEURAL NETWORK ARCHITECTURES

The neural network architecture of the PPO agent used in our experiments is shown in Figure 7. The PPO agent comprises actor and critic modules, which share the same feature extraction layers.

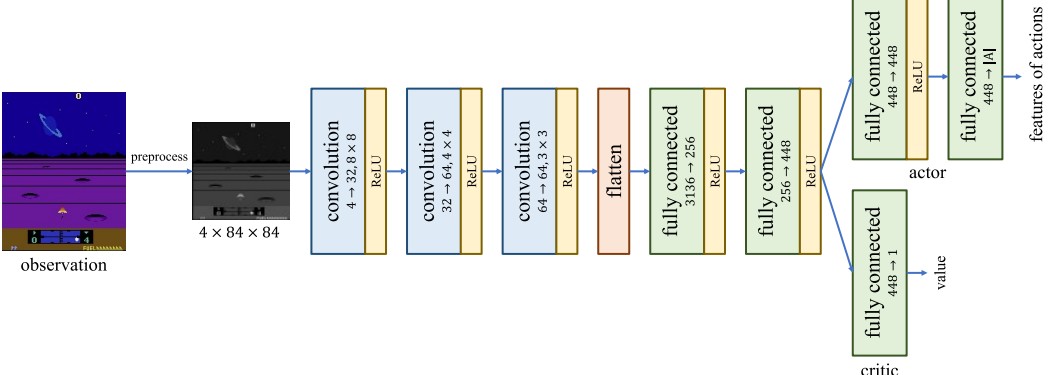

Figure 7: The neural network architecture of the PPO agent in our experiments.

For the random networks that map one frame of preprocessed observation to a 512-length feature vector, the architecture is depicted in Figure 8.

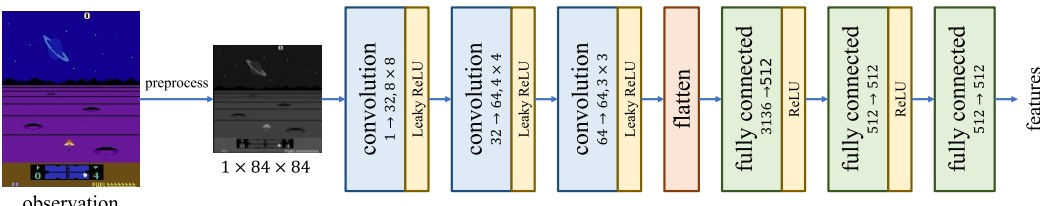

Figure 8: The neural network architecture of the random network in our experiments.

### A.2.4 HARDWARE CONFIGURATIONS

The experiments are conducted on machines mainly with two kinds of configurations:

1. The GPU is NVIDIA Tesla A100 with 40GB memory. The CPU is Intel Xeon Gold 6326 with 16 cores and 32 threads.

2. The GPU is NVIDIA Tesla H100 with 40GB memory. The CPU is AMD Epyc 9334 with 32 cores and 64 threads.

The experiments are implemented by *PyTorch* in version 2.0.1 and *CUDA* in version 11.7.

### A.3 ADDITIONAL EXPERIMENTAL RESULTS

To support the main results of the ablation study in our paper, we provide quantified results of DuRND with a single type of reward in Table 5. The results show that both types of rewards are essential for the DuRND framework to achieve the best performance.

Table 5: Ablation study: the average episodic returns with standard errors of DuRND with a single type of reward

| Environments | DuRND | DuRND with only $R^{nov}$ | DuRND with only $R^{con}$ |
|---|---|---|---|
| *Freeway* | $23.22 \pm 0.01$ | $19.63 \pm 0.01$ | $15.57 \pm 0.01$ |
| *Frogger* | $14.36 \pm 0.00$ | $10.10 \pm 0.00$ | $10.81 \pm 0.00$ |
| *Solaris* | $18.91 \pm 0.02$ | $7.83 \pm 0.01$ | $17.61 \pm 0.01$ |
| *BeamRider* | $18.05 \pm 0.01$ | $9.45 \pm 0.00$ | $13.07 \pm 0.01$ |
| *DefendLine* | $8.52 \pm 0.00$ | $2.65 \pm 0.00$ | $4.09 \pm 0.00$ |
| *SaveCenter* | $6.33 \pm 0.00$ | $3.08 \pm 0.00$ | $4.31 \pm 0.00$ |
| *CollectKit* | $20.87 \pm 0.01$ | $11.12 \pm 0.01$ | $9.11 \pm 0.01$ |
| *SlayGhosts* | $15.60 \pm 0.00$ | $7.22 \pm 0.00$ | $4.03 \pm 0.00$ |
| *ThreeRooms* | $0.86 \pm 0.00$ | $0.26 \pm 0.00$ | $0.06 \pm 0.00$ |
| *TMaze* | $0.96 \pm 0.00$ | $0.93 \pm 0.00$ | $0.52 \pm 0.00$ |

