# OpenReview forum: "DuRND: Rewarding from Novelty to Contribution for Reinforcement Learning via Dual Random Networks Distillation"
_ICLR.cc/2025/Conference — Submitted to ICLR 2025_

### Official Review · Reviewer_Ktvw · 2024-10-25

**Soundness:** 2
**Presentation:** 3
**Contribution:** 2
**Rating:** 3
**Confidence:** 3

**Summary:**

The paper's method is established on the observation that novelty-based reward can divert agents from their main objects and value-based reward lacks sufficient early exploration. The authors then propose a framework (DuRND) integrating 2 groups of lightweight random network pairs that jointly generate novelty and contribution rewards. To balance exploitation and exploration during training, DuRND scales the coefficients for the novelty and contribution rewards throughout the learning process. Finally, the authors integrated DuRND into PPO.

I think the problem this paper tries to solve is important, but some clarifications are needed to help me gauge the contribution of this work.

**Strengths:**

- Learning in sparse reward tasks is a long-standing problem in RL, the paper is taking on an important challenge.
- The paper's idea is simple and clearly presented.

**Weaknesses:**

- It is unclear how challenging the evaluated tasks are. Unlike Montezuma's Revenge, a notoriously hard to solve problem, it is unclear how challenging the tasks in which DuRND is evaluated and thus it is hard to gauge the contribution. For example, the [highest score achieved in Freeway is 34](https://github.com/cshenton/atari-leaderboard) while the best algo in Figure 6 achieves 25.
- Somewhat contradictory findings. While in the abstract and the introduction, the authors say that *"The former [novelty-based reward] encourages agents to explore less visited areas but can divert them from their main objectives, while the latter [value-based reward] promotes stable late-stage convergence but often lacks sufficient early exploration."*, Figure 6 shows that novelty reward can actually achieve descent scores (e.g., Freeway, Frogger and TMaze). This questions the paper's assumption.

**Questions:**

1. Can the authors provide scores from other model-free RL algos on the tasks? Like PPO? This would also allow a reader to compare the vanilla PPO with DuRND, which is a modified version of PPO in this paper.
2. Is the 2nd point in the Weaknesses section reasonable to the authors?
3. When the reward is sparse, there are few success trajectories. Does it cause a problem for learning the Success RN module? How did you overcome this problem?
4. Is $T_{max}$ for each task tuned? I don't see it in table 4. If so, did you also tune the HPs for the baselines?
5. I find the toy task to be comprehensive and a good tool to understand DuRND, how are the states represented? One-hot encoding? Can the authors provide code (it's not in the current supplementary material)? Both training and the learned model would be appreciated.

---

> ### Author Response · Authors · 2024-11-23
>
> ***Response to Reviewer Ktvw Part 1/2***
>
> Dear reviewer,
>
> Thank you for your comments and we address your concerns below:
>
> # Weaknesses
>
> > * It is unclear how challenging the evaluated tasks are. Unlike Montezuma's Revenge, a notoriously hard to solve problem, it is unclear how challenging the tasks in which DuRND is evaluated and thus it is hard to gauge the contribution. For example, the highest score achieved in Freeway is 34 while the best algo in Figure 6 achieves 25.
>
> The selected tasks are classic RL benchmark tasks, including *Atari games*, *VizDoom games*, and *3D maze*. To ensure high difficulty, we use extremely sparse-reward settings, making these tasks highly challenging. DuRND consistently outperforms 6 representative reward shaping baselines across 10 environments, which strongly demonstrates its effectiveness.
>
> Regarding the performance in *Freeway*, we modified the original reward structure to make it more challenging. Specifically, we removed bonus rewards and only awarded a value of $1$ for successfully crossing the road. Therefore, the reward structure and the highest reward one agent can achieve differ from the original game. Details about this modified reward model are listed in the *Appendix A.1*.
>
> > * Somewhat contradictory findings. While in the abstract and the introduction, the authors say that "The former [novelty-based reward] encourages agents to explore less visited areas but can divert them from their main objectives, while the latter [value-based reward] promotes stable late-stage convergence but often lacks sufficient early exploration.", Figure 6 shows that novelty reward can actually achieve descent scores (e.g., Freeway, Frogger and TMaze). This questions the paper's assumption.
>
> Our results in Figure 6 is not a contradiction but well supports and aligns with our claims in the abstract and introduction. Figure 6 presents an ablation study to analyze the roles of different reward modules (*novelty reward* and *contribution reward*). The results show that *DuRND with only novelty reward* achieves decent scores in some environments, such as Freeway, Frogger, and TMaze. This aligns with our statement: “The novelty-based reward encourages agents to explore less visited areas but can divert them from their main objectives,” thus may achieve decent scores in the later stages.
>
> The ablation study highlights that, in the absence of contribution rewards, the agent may focus excessively on exploring novel but suboptimal states, resulting in difficulty recovering in later stages. These findings demonstrate two key points:
> 1. The *contribution reward* plays a crucial role in improving the overall performance of DuRND.
> 2. Agents relying solely on novelty rewards may deviate from their main objectives.

---

> > ### Author Response · Authors · 2024-11-23
> >
> > ***Response to Reviewer Ktvw Part 2/2***
> >
> > # Questions
> >
> > > 1. Can the authors provide scores from other model-free RL algos on the tasks? Like PPO? This would also allow a reader to compare the vanilla PPO with DuRND, which is a modified version of PPO in this paper.
> >
> > Regarding the comparsion of PPO, initially, as PPO is the backbone algorithm for the RND baseline, and previous work has demonstrated that RND outperforms vanilla PPO, we focused on comparisons with RND in our experiments. However, we understand the importance to compare with this backbone algorithm, so we conducted experiments with vanilla PPO, and the results are shown in the table below:
> >
> > | Algo. | Freeway | Frogger | Solaris | BeamRider | DefendLine | SaveCenter | CollectKit | SlayGhosts | ThreeRooms | TMaze|
> > | :---: | :---: | :---: | :---: | :---: | :---: | :---: | :---: | :---: | :---: | :---: |
> > | DuRND | 23.22 $\pm$ 0.01 | 14.36 $\pm$ 0.00 | 18.91 $\pm$ 0.02 | 18.05 $\pm$ 0.01 | 8.52 $\pm$ 0.00 | 6.33 $\pm$ 0.00 | 20.87 $\pm$ 0.01 | 15.60 $\pm$ 0.00 | 0.86 $\pm$ 0.00 | 0.96 $\pm$ 0.00 |
> > | RND | 14.77 $\pm$ 0.01 | 8.59 $\pm$ 0.00 | 6.07 $\pm$ 0.00 | 11.96 $\pm$ 0.00 | 1.11 $\pm$ 0.00 | 2.37 $\pm$ 0.00 | 14.59 $\pm$ 0.01 | 10.18 $\pm$ 0.00 | 0.00 $\pm$ 0.00 | 0.97 $\pm$ 0.00 |
> > | PPO | 10.67 $\pm$ 0.00 | 3.25 $\pm$ 0.00 | 1.82 $\pm$ 0.01 | 10.23 $\pm$ 0.00 | 0.00 $\pm$ 0.00 | 0.00 $\pm$ 0.00 | 5.89 $\pm$ 0.00 | 8.15 $\pm$ 0.02 | 0.00 $\pm$ 0.00 | 0.94 $\pm$ 0.00 |
> >
> > > 2. Is the 2nd point in the Weaknesses section reasonable to the authors?
> >
> > The results in Figure 6 are not contradictory to our conclusions, instead, they support and align with our claims. Please see our response above under *Weaknesses 2* for detailed explanation.
> >
> > > 3. When the reward is sparse, there are few success trajectories. Does it cause a problem for learning the Success RN module? How did you overcome this problem?
> >
> > In the early stages of training, the sparse rewards will result in fewer success states, however, this does not affect the early exploration stage. Because in the early stages, the agent is mainly encouraged to explore novel states by the dominance of the novelty reward. And the novelty reward considers the novelty of states in both success and failure trajectories ($R^{novel}(s) = e_S(s) + e_F(s)$), ensuring the agent is encouraged to explore novel states regardless of the number of success trajectories. Over time, this exploration naturally leads to learning success trajectories.
> >
> > > 4. Is $T_{max}$ for each task tuned? I don't see it in table 4. If so, did you also tune the HPs for the baselines?
> >
> > While defining $T_{\text{max}}$ may require some environment-specific knowledge, such as reward sparsity, we found that setting $T_{\text{max}} = \frac{1}{4} \times T_{\text{episode}}$ works consistently well across all environments in our experiments (except for *ThreeRooms* and *TMaze*, where we use the full trajectory). This heuristic divides an episode into four sub-trajectories, with the contribution of each sub-trajectory determined by whether it achieves a positive reward within the segment.
> >
> > Regarding the hyperparameters for the baselines, we consistently applied the default settings provided in their respective papers and implementations. The code sources for the baselines are as follows:
> >
> > 1. The [CleanRL library](https://github.com/vwxyzjn/cleanrl) for implementing *RND* and *PPO*.
> > 2. The [RLeXplore library](https://github.com/RLE-Foundation/RLeXplore) for implementing *#Explo* and *ROSA*.
> > 3. The official code provided in the original papers for [ExploRS](https://github.com/machine-teaching-group/neurips2022_exploration-guided-reward-shaping), [ReLara](https://github.com/mahaozhe/ReLara), and [SORS](https://github.com/hiwonjoon/IROS2021_SORS).
> >
> > > 5. I find the toy task to be comprehensive and a good tool to understand DuRND, how are the states represented? One-hot encoding? Can the authors provide code (it's not in the current supplementary material)? Both training and the learned model would be appreciated.
> >
> > Yes, the states in the toy task are represented using one-hot encoding. The code for both training and the learned model will be made publicly available on GitHub after the review period.
> >
> > Once again, we thank for your comments and hope our responses address your concerns.

---

> ### Comment · Reviewer_Ktvw · 2024-11-25
>
> Thank you for the responses.
>
> > Yes, the states in the toy task are represented using one-hot encoding. The code for both training and the learned model will be made publicly available on GitHub after the review period.
>
> You provided the code for the RL tasks, why delaying the code submission for this toy task?
>
> Due to the concern above, I'll keep my current rating.

---

> > ### Author Response · Authors · 2024-11-25
> >
> > The toy task is a chain task with 31 states, encoding the state as a one-hot vector, which is quite straightforward and simple. Here is the code for the Chain task:
> >
> >
> > ```
> > import numpy as np
> > from gymnasium import Env, spaces
> >
> > class ChainEnv(Env):
> >     def __init__(self):
> >         super().__init__()
> >         self.cur_state = 15
> >         self.cur_time = 0
> >         self.max_states = 31
> >         self.max_time = 20
> >
> >         # Action space: 0 (left), 1 (stay), 2 (right)
> >         self.action_space = spaces.Discrete(3)
> >
> >         self.observation_space = spaces.Box(low=0, high=1, shape=(self.max_states,), dtype=np.float32)
> >
> >     def reset(self, seed=None, options=None):
> >         super().reset(seed=seed)
> >         self.cur_state = 15
> >         self.cur_time = 0
> >
> >         state = np.zeros(self.max_states, dtype=np.float32)
> >         state[self.cur_state] = 1.0
> >         return state, {}
> >
> >     def step(self, action):
> >         self.cur_time += 1
> >
> >         if action == 0:
> >             # action: left
> >             self.cur_state = max(self.cur_state - 1, 0)
> >         elif action == 2:
> >             # action: right
> >             self.cur_state = min(self.cur_state + 1, 30)
> >         elif action == 1:
> >             # action: stay
> >             pass
> >         else:
> >             raise ValueError("Invalid action")
> >
> >         # Calculate reward
> >         reward = 1 if self.cur_state == 30 else 0
> >
> >         # Check if the episode is done
> >         done = self.cur_time >= self.max_time or reward == 1
> >
> >         # One-hot encode the current state
> >         state = np.zeros(self.max_states, dtype=np.float32)
> >         state[self.cur_state] = 1.0
> >
> >         return state, reward, done, False, {}
> > ```

---

> > > ### Comment · Reviewer_Ktvw · 2024-11-25
> > >
> > > Please submit runnable code for training and test, as was requested in my first review comment.

---

> > > > ### Author Response · Authors · 2024-11-25
> > > >
> > > > The training code has been provided in our "Supplementary Materials", which can run the `ChainEnv` environment provided above.

---

### Official Review · Reviewer_8cSS · 2024-10-26

**Soundness:** 1
**Presentation:** 3
**Contribution:** 2
**Rating:** 3
**Confidence:** 4

**Summary:**

The paper proposes a novel algorithm DuRND, a refined version of RND (Random Network Distillation), by categorizing the states into successful states and failed states. This allows the agent to obtain a more refined intrinsic reward structure.

**Strengths:**

1. The method is easy to understand.
2. Presumably easy to implement from an existing RND implementation.
3*. The results seem to be strong, but I have a number of concerns, which I will elaborate on in the weaknesses section.

**Weaknesses:**

The idea of splitting the successful and failed states is interesting, but not convincing. My understanding is that the authors try to prevent the exploration algorithms to over-explore by re-weighting the reward based on the success/failure prior, so that the agent would be more exploratory towards successful states, and less towards failed states. However, this can be simply done by computing the intrinsic reward conditional on value [1], which is a more general indicator of the quality of current state.

The proposed algorithm integrates the intuition mentioned above, but significantly increases the complexity of the algorithm compared to RND, araising some of my concerns discussed below.

Despite the results seem strong, there are several concerns raise in terms of the experiment:
1. PPO is missing in the comparison, which is very crucial for experiment of any PPO-based algorithms.

2. The proposed algorithm DuRND introduces two new hyperparameters $\omega$ and $\lambda$ into the formulation, which does not exist in RND. In the original RND experiment, the coefficient of the intrinsic reward is effectively fixed to 0.5 given by $2A_{env} + A_{rnd}$. However, the authors mentioned they gradually decrease $\lambda$ from 1 to 0 and increase $\omega$ from 0 to 1, which drastically different from the choice of RND. I will elaborate on this point:
* RL algorithms are \textbf{VERY} sensitive to the choice of the coefficient like $\omega$ and $\lambda$ in this paper. Hence it is very concerning whether the performance improvement mostly coming from the choice of the hyperparameter. The fact that RL algorithm would prefer a decreasing novelty reward coefficient as the training proceeds, so that the algorithm can start to emphasize exploitation sooner, hence achieves better sample efficiency. I would potentially lift my rating if the authors can show that DuRND can still out-perform other baseline algorithms with same schedule of coefficients.
* Similar to the previous point, it is also crucial to control the speed of the converge rate of the intrinsic reward model, i.e. the learning rate of $f(\dot ; \theta)$. This is not mentioned in the paper.

3. The proposed algorithm DuRND requires a way to determining whether a trajectory is successful or failed. The authors mention that they use the $\sum_{step=1}^{T_{max}} r \geq 1$ condition, to determine whether a sub-trajectory rollout in the current iteration is successful or not. This introduces two problems:
* $T_{max}$ is still a hyperparameter, the value of which is not provided in the paper.
* Intuitively, the suitable choice of such hyperparameter would correlate to the periodicity of the environment. Consider a episodic setting, if the task is a navigation task, like TMaze and ThreeRooms, such $T_{max}$ should be equal to the horizon $H=500$, whereas in the environment like Solaris, the $T_{max}$ should presumably be smaller, otherwise many bad states would be considered as successful state. Hence it would be very difficult to tune when the periodicity of the environment is very hard to know, for example, legged robotics tasks.

Overall, I am not convinced that the improvement of the performance solely coming from the characterization of the successful/failed states, during the design process, the authors introduced three hyperparameters that may not be chosen in a systematic or unified way across all types of environment.

[1] Accelerating Reinforcement Learning withValue-Conditional State Entropy Exploration

**Questions:**

1. What is the performance of PPO in this set of environment?
2. What is the performance of algorithms in the set of environment with original reward structure?
3. What is the choice of hyperparameters $\lambda$, the coefficient of intrinsic reward of baseline algorithms?
4. What is the performance of RND, as well as other baseline algorithms, when you also decrease their coefficient $\lambda$ from 1 to 0 linearly?
5. What is the choice of $T_{max}$ used in different environment? How do you choose the $T_{max}$?

---

> ### Author Response · Authors · 2024-11-23
>
> Dear reviewer,
>
> Thanks a lot for your insightful feedback. We want to address your concerns below:
>
> ## Comparison with PPO
>
> > (Weakness) 1. PPO is missing in the comparison, which is very crucial for experiment of any PPO-based algorithms.
> > (Question) 1. What is the performance of PPO in this set of environment?
>
> *(Weakness 1 and Question 1)* Regarding the comparison of PPO, initially, as the PPO is the backbone algorithm for the RND baseline, and the RND has outperformed the vanilla PPO in previous works, so we only include the comparison with RND in our experiments. However, we understand the importance of comparing with this backbone algorithm, so we conducted experiments with vanilla PPO, and the results are shown in the table below:
>
>
> | Algo. | Freeway | Frogger | Solaris | BeamRider | DefendLine | SaveCenter | CollectKit | SlayGhosts | ThreeRooms | TMaze|
> | :---: | :---: | :---: | :---: | :---: | :---: | :---: | :---: | :---: | :---: | :---: |
> | DuRND | 23.22 $\pm$ 0.01 | 14.36 $\pm$ 0.00 | 18.91 $\pm$ 0.02 | 18.05 $\pm$ 0.01 | 8.52 $\pm$ 0.00 | 6.33 $\pm$ 0.00 | 20.87 $\pm$ 0.01 | 15.60 $\pm$ 0.00 | 0.86 $\pm$ 0.00 | 0.96 $\pm$ 0.00 |
> | RND | 14.77 $\pm$ 0.01 | 8.59 $\pm$ 0.00 | 6.07 $\pm$ 0.00 | 11.96 $\pm$ 0.00 | 1.11 $\pm$ 0.00 | 2.37 $\pm$ 0.00 | 14.59 $\pm$ 0.01 | 10.18 $\pm$ 0.00 | 0.00 $\pm$ 0.00 | 0.97 $\pm$ 0.00 |
> | PPO | 10.67 $\pm$ 0.00 | 3.25 $\pm$ 0.00 | 1.82 $\pm$ 0.01 | 10.23 $\pm$ 0.00 | 0.00 $\pm$ 0.00 | 0.00 $\pm$ 0.00 | 5.89 $\pm$ 0.00 | 8.15 $\pm$ 0.02 | 0.00 $\pm$ 0.00 | 0.94 $\pm$ 0.00 |
>
> ## Reward Weighting and Scaling
>
> > (weakness) 2. The proposed algorithm DuRND introduces two new hyperparameters w and $\lambda$ into the formulation, which does not exist in RND. In the original RND experiment, the coefficient of the intrinsic reward is effectively fixed to 0.5 given by $2A_{env} + A_{rnd}$. However, the authors mentioned they gradually decrease $\lambda$ from 1 to 0 and increase $w$ from 0 to 1, which drastically different from the choice of RND. I will elaborate on this point:
> >    * RL algorithms are \textbf{VERY} sensitive to the choice of the coefficient like w and $\lambda$ in this paper. Hence it is very concerning whether the performance improvement mostly comes from the choice of the hyperparameter. The fact that RL algorithm would prefer a decreasing novelty reward coefficient as the training proceeds, so that the algorithm can start to emphasize exploitation sooner, hence achieves better sample efficiency. I would potentially lift my rating if the authors can show that DuRND can still out-perform other baseline algorithms with same schedule of coefficients.
> >    * Similar to the previous point, it is also crucial to control the speed of the converge rate of the intrinsic reward model, i.e. the learning rate of f. This is not mentioned in the paper.
>
> *(Weakness 2 and Question 4)* **Regarding the hyperparameters** $\lambda$ and $\omega$, which control the weighting of the shaped rewards, the linear schedule primarily follows the idea behind the $\epsilon$-greedy strategy, which decays $\epsilon$ linearly to balance exploration and exploitation over time. The goal is to let different rewards dominate at different training stages: during the early stages, the *novelty reward* promotes exploration, while in the later stages, the *contribution reward* encourages exploitation.
>
> Regarding the robustness of DuRND to the choice of $\lambda$ and $\omega$, our experiments show that DuRND consistently outperforms the baselines across all environments using the same linear schedule ($1 \rightarrow 0$ for $\lambda$ and $0 \rightarrow 1$ for $\omega). This demonstrates that DuRND is not sensitive to the choice or schedule of these coefficients.
>
> Regarding the learning rate of the random network (RN) modules, we indeed use a relatively low learning rate to ensure the RNs do not converge too quickly. This helps maintain sufficient distinction among visited states throughout training. We will include this detail in the revised manuscript to address this omission.
>
> Regarding the concern about whether "DuRND can still outperform other baseline algorithms with the same schedule of coefficients," and the related question:
>
> > (Question) 4. What is the performance of RND, as well as other baseline algorithms, when you also decrease their coefficient $\lambda$ from 1 to 0 linearly?
>
> We understand this is an important point and worth investigating. Thus we conducted additional experiments on RND with the same linear schedule for the novelty reward coefficient as our DuRND. The results are shown in the table below:

---

> > ### Author Response · Authors · 2024-11-23
> >
> > ***Response to Reviewer 8cSS Part 2/3***
> >
> >
> > | Algo. | Freeway | Frogger | Solaris | BeamRider | DefendLine | SaveCenter | CollectKit | SlayGhosts | ThreeRooms | TMaze|
> > | :---: | :---: | :---: | :---: | :---: | :---: | :---: | :---: | :---: | :---: | :---: |
> > | DuRND | 23.22 $\pm$ 0.01 | 14.36 $\pm$ 0.00 | 18.91 $\pm$ 0.02 | 18.05 $\pm$ 0.01 | 8.52 $\pm$ 0.00 | 6.33 $\pm$ 0.00 | 20.87 $\pm$ 0.01 | 15.60 $\pm$ 0.00 | 0.86 $\pm$ 0.00 | 0.96 $\pm$ 0.00 |
> > | DuRND with only $R^{nov}$ | 19.63 $\pm$ 0.01 | 10.10 $\pm$ 0.00 | 7.83 $\pm$ 0.01 | 9.45 $\pm$ 0.00 | 2.65 $\pm$ 0.00 | 3.08 $\pm$ 0.00 | 11.12 $\pm$ 0.01 | 7.22 $\pm$ 0.00 | 0.26 $\pm$ 0.00 | 0.93 $\pm$ 0.00 |
> > | RND | 14.77 $\pm$ 0.01 | 8.59 $\pm$ 0.00 | 6.07 $\pm$ 0.00 | 11.96 $\pm$ 0.00 | 1.11 $\pm$ 0.00 | 2.37 $\pm$ 0.00 | 14.59 $\pm$ 0.01 | 10.18 $\pm$ 0.00 | 0.00 $\pm$ 0.00 | 0.97 $\pm$ 0.00 |
> > | RND (linearly decreasing $\lambda$) | 13.68 $\pm$ 0.00 | 10.12 $\pm$ 0.00 | 3.46 $\pm$ 0.01 | 14.85 $\pm$ 0.00 | 3.73 $\pm$ 0.01 | 2.04 $\pm$ 0.02 | 11.22 $\pm$ 0.00 | 8.93 $\pm$ 0.00 | 0.00 $\pm$ 0.00 | 0.97 $\pm$ 0.00 |
> >
> > **Analysis on the Results**
> >
> > We observe two key findings from the results, which we analyze as follows:
> >
> > 1. **RND with the same linear decreasing schedule for the novelty reward coefficient achieves similar or even worse performance compared to the original RND.**
> >
> > This can be attributed to the intrinsic nature of RND's novelty reward. In standard RND, the novelty reward scale naturally decreases over training because most states are eventually visited, causing the novelty reward to decrease. Adding a linear decreasing schedule on top of this could accelerate the reduction of the novelty reward scale. While this creates longer exploitation, it also leads the agent to stop exploring prematurely, resulting in performance degradation. However, in some tasks, such as the *Freeway*, this approach shows slight improvement. This is likely because sufficient exploration in the early stages already allows the agent to discover the majority of positive states. In such cases, halting exploration earlier could indeed be beneficial.
> >
> > 2. **RND with the same linear decreasing schedule for the novelty reward coefficient fails to outperform DuRND and even underperforms compared to DuRND with only $R^{nov}$.**
> >
> > This result can be explained by two main factors:
> >
> > **(a)** Why RND with linear decreasing novelty reward does not outperform DuRND with only $R^{nov}$:
> >
> > DuRND categorizes states into two scenarios, recorded by the success RN module and the failure RN module, respectively. The novelty reward in DuRND is determined by the sum of the errors from these two modules, $e_S(s) + e_F(s)$. This enables DuRND to create three levels of novelty bonuses:
> >
> > - **High novelty**: The state is unseen in both success and failure trajectories, requiring more exploration.
> > - **Medium novelty**: The state is seen in failure trajectories but not in success trajectories, warranting some exploration since states in failure trajectories might eventually lead to success.
> > - **Low novelty**: The state is seen in both success and failure trajectories, signaling that exploration can stop.
> >
> > During the early exploration phase, most states are categorized as failures and stored in the failure RN module. As a result, $e_F(s)$ is relatively low, while $e_S(s)$ remains high. **The "success RN module" will "drag" the novelty reward to avoid it from decreasing too quickly**, thereby prolonging the effectiveness of rewarding novelty. Over time, the novelty reward decays due to the decreasing $\lambda$ or the diminishing novelty itself.
> >
> > This means we are intentionally using an "AND" condition to determine the novelty bonus, which means that the state can be considered novel no matter whether it is novel in the success module or the failure module. This is because we want to encourage exploration in states that are novel in either module. In contrast, RND records all states in a single module, regardless of whether they are successes or failures. Consequently, novelty rewards for many states decrease quickly in the early stages. Combined with the linear decay of $\lambda$, this can lead to premature termination of exploration, resulting in poorer performance.
> >
> > **(b)** Why RND with linear decreasing novelty reward does not outperform full DuRND:
> >
> > This highlights the crucial role of the *contribution reward* in DuRND. After sufficient exploration, DuRND effectively distinguishes states that are more likely to lead to positive rewards, enabling better exploitation. This further demonstrates that our contribution reward plays a critical role in superior performance.

---

> ### Author Response · Authors · 2024-11-23
>
> ***Response to Reviewer 8cSS Part 3/3***
>
> ## Success and Failure Definition and $T_{max}$
>
> > (weakness) 3. The proposed algorithm DuRND requires a way to determining whether a trajectory is successful or failed. The authors mention that they use the condition, to determine whether a sub-trajectory rollout in the current iteration is successful or not. This introduces two problems:
> >    * is still a hyperparameter, the value of which is not provided in the paper.
> >    * Intuitively, the suitable choice of such hyperparameter would correlate to the periodicity of the environment. Consider a episodic setting, if the task is a navigation task, like TMaze and ThreeRooms, such Tmax should be equal to the horizon H=500, whereas in the environment like Solaris, the Tmax should presumably be smaller, otherwise many bad states would be considered as successful state. Hence it would be very difficult to tune when the periodicity of the environment is very hard to know, for example, legged robotics tasks.
>
> > (Question) 5. What is the choice of Tmax used in different environment? How do you choose the Tmax?
>
> *(Weakness 3 and Question 5)* **Regarding the success and failure definition**, our intention is to define a metric that evaluates whether a trajectory (or sub-trajectory) contributes to obtaining a positive reward, rather than strictly achieving a goal. The basic assumption is that some states are far from obtaining a positive reward, while others can more directly lead to positive rewards. The latter indicates a higher contribution and thus deserves a higher reward. While we are inspired by the concept of success in goal-achieving environments, we have extended it, which is why we instead refer to this reward as a **contribution reward**, as it quantifies the contribution/importance of states leading to positive rewards. More importantly, because our work is targeted at sparse-reward environments, in such environments, getting a very rare positive reward is itself a clear and unambiguous way of determining if a sub-trajectory is successful or not (i.e., referred to milestone in our paper). (The reward structure of the environments is detailed in the *Appendix A.1*.)
>
> Regarding the $T_{max}$, while defining the hyperparameter $T_{max}$ may require some environment-specific knowledge, like how sparse the rewards are, we have observed that setting $T_{max} = \frac{1}{4} \times T_{episode}$ consistently yields strong results across all environments in our experiments (except the ThreeRooms and TMaze, where we use the whole trajectory). This heuristic divides an episode into four sub-trajectories, and the contribution of a sub-trajectory is determined by whether it achieves a positive reward within this segment. Although this heuristic involves some approximation, its effectiveness has been empirically validated in our paper.
>
> ## Performance with Original Reward Structure
>
> > (Question) 2. What is the performance of algorithms in the set of environment with original reward structure?
>
> We would like to highlight that **all of the experiments in our paper are reporting the performance with the original environmental reward structure**, rather than the shaped reward structure. As although all reward shaping methods introduce additional rewards, but still the original environmental reward is the only objective that the agent is trying to optimize. The shaped rewards serve merely as auxiliary signals to facilitate learning, but they should not interfere with the optimization of the original reward.
>
> ## Choice of Hyperparameters in Baselines
>
> > (Question) 3. What is the choice of hyperparameters $\lambda$, the coefficient of intrinsic reward of baseline algorithms?
>
> Regarding the coefficient of the shaped rewards in the baselines, we consistently applied the default settings as specified in their respective papers and implementations. Each algorithm is studied with its own proposed method for setting the coefficients of the shaped rewards, and we did not alter these default settings. The codes for the baselines are from:
>
> 1. The [CleanRL library](https://github.com/vwxyzjn/cleanrl) for implementing *RND* and *PPO*.
> 2. The [RLeXplore library](https://github.com/RLE-Foundation/RLeXplore) for implementing *#Explo* and *ROSA*.
> 3. The official code provided in the original papers for [ExploRS](https://github.com/machine-teaching-group/neurips2022_exploration-guided-reward-shaping), [ReLara](https://github.com/mahaozhe/ReLara), and [SORS](https://github.com/hiwonjoon/IROS2021_SORS).
>
> Once again, we appreciate your comments and hope that our responses address your concerns.

---

> > ### Comment · Reviewer_8cSS · 2024-11-25
> >
> > I would like to thank the authors for providing explanations for my questions and concerns and I will keep my current rating.
> > The whole algorithm introduces too many new hyperparameters, without a systematic way of tuning them. And my concern for the experiment still remains, and as Reviewer Ktvw mentioned, the exploration difficulty of the environments of choice is questionable.

---

> ### Author Response · Authors · 2024-11-25
>
> Thanks a lot for your feedback.
>
> Regarding the hyperparameters introduced, in our newly posted **Overall Response**, we demonstrated that the **DuRND without the scheduling** achieves almost the same performance as the original DuRND across all environments. This indicates that the linear scheduling of $\lambda$ and $\omega$ has minimal impact on DuRND's performance, and removing the scheduling operation can simplify the DuRND framework significantly. We hope these new results can help clarify the concerns and provide a more comprehensive understanding of DuRND.

---

### Official Review · Reviewer_QctF · 2024-10-28

**Soundness:** 3
**Presentation:** 3
**Contribution:** 2
**Rating:** 6
**Confidence:** 4

**Summary:**

Random network distillation is an unsupervised technique designed to enhance the exploratory behavior of an agent by fitting a predictor network to the latent outputs of a randomly initialized but fixed target network. The deviation between target and predictor, commonly measured by the mean squared error, is typically higher in underexplored areas of the search space, translating into a higher auxiliary "reward" signal that encourages exploration. The paper "DuRND: Rewarding from Novelty to Contribution for Reinforcement Learning via Dual Random Networks Distillation" proposes an extension to classical RND by introducing two distinct random network modules—one for states deemed "successful" and another for states associated with "failure." This innovation allows for the derivation of both a "novelty" and a "contribution" reward signal, striking a balance between exploratory and exploitative behavior. The proposed method is evaluated on three benchmark environments: Atari, VizDoom, and MiniWorld.

**Strengths:**

•	The paper is well-written, structured, and easy to follow.

•	The exploration vs. exploitation dilemma remains a core issue in reinforcement learning.

•	The general approach is elegantly simple and lightweight, meaning it can be easily integrated into any online reinforcement learning algorithm without significant overhead.

•	The authors provide a thorough experimental evaluation, reporting results from over 10 different training runs across 10 distinct environments, and comparing DuRND to six baseline methods.

•	The extensive evaluation demonstrates the utility of both the "novelty" and "contribution" reward signals in isolation, and shows their synergy when used together.

**Weaknesses:**

•	Despite the extensive evaluation in the experiments section, I have a significant concern, which the authors have acknowledged but deferred to future work: the weighting of the shaped reward signals. The paper aims to improve handling of the exploration-exploitation dilemma, and while it achieves this, the shaped rewards rely on additional hyperparameters—λ and ω—which are linearly adjusted in the current implementation. I believe this aspect warrants more in-depth discussion and empirical analysis within the paper itself.

•	In addition, the method relies on success and failure labels to update the respective network modules, but in ambiguous or multi-objective tasks, defining success and failure may not be straightforward.

•	While the additional novelty introduced by DuRND is incremental compared to classical RND, I still believe the contribution is valuable and fills a gap in the current literature.

**Questions:**

•	How should the λ and ω hyperparameters be set in general, and more critically, how can they be set in the absence of prior knowledge about the scaling of the environmental reward (r_env)?

•	The linear scaling approach requires an end point; how should this be determined without resorting to expensive and time-consuming experimental tuning?

•	How robust is the performance of DuRND if the weight scheduler is suboptimal? This question is critical, as real-world applications often cannot afford perfect tuning of hyperparameters, and the performance may degrade substantially if these are not set optimally.

Given the importance of these issues, I feel they should not be relegated to the future work section. Instead, this discussion should be incorporated into the main body of the paper, potentially reducing sections 4.3 and 5.2 to make space for this analysis.

--
Post-rebuttal: After reading all reviews and rebuttals, it is evident that there is a significant overlap in the issues found. While I appreciate the clarifications provided in the rebuttal, the most critical aspect -- a more detailed analysis of λ and ω, including their scaling with respect to environmental rewards, the endpoint of linear scaling, and suboptimal weight scheduling -- remains largely unaddressed. Therefore, I will maintain my current score.

---

> ### Author Response · Authors · 2024-11-23
>
> Dear reviewer,
>
> Thank you very much for your valuable feedback. We address your concerns in the following sections:
>
> ## Reward Weighting
>
> > * (Question 1) How should the $\lambda$ and $\omega$ hyperparameters be set in general, and more critically, how can they be set in the absence of prior knowledge about the scaling of the environmental reward (r_env)?
>
> *(Question 1)* **Regarding the hyperparameters** $\lambda$ and $\omega$, which control the weighting of the shaped rewards, their values are indeed closely related to the scaling of the environmental rewards ($r^{env}$). Balancing the shaped rewards with the original environmental rewards is a common challenge for all reward shaping methods, and they typically require knowledge of the environmental reward scale. However, we believe that this information is often readily available. In the current literature on reward shaping (e.g., RND, ExploRS, ReLara, SORS, etc.), a common practice is to set the shaped reward scale to approximately $0.5$ times the scale of the environmental reward. This heuristic has been shown to work well in a variety of settings and serves as a practical guideline for choosing these hyperparameters.
>
> ## Reward Scaling
>
> > * (weakness 1) Despite the extensive evaluation in the experiments section, I have a significant concern, which the authors have acknowledged but deferred to future work: the weighting of the shaped reward signals. The paper aims to improve handling of the exploration-exploitation dilemma, and while it achieves this, the shaped rewards rely on additional hyperparameters—$\lambda$ and $\omega$—which are linearly adjusted in the current implementation. I believe this aspect warrants more in-depth discussion and empirical analysis within the paper itself.
> > * (Question 2) The linear scaling approach requires an end point; how should this be determined without resorting to expensive and time-consuming experimental tuning?
> > * (Question 3) How robust is the performance of DuRND if the weight scheduler is suboptimal? This question is critical, as real-world applications often cannot afford perfect tuning of hyperparameters, and the performance may degrade substantially if these are not set optimally.
>
> *(Weakness 1 and Question 2, 3)* **Regarding the reward weights scheduling**, our approach is conceptually similar to the well-known $\epsilon$-greedy strategy, which linearly decays to balance exploration and exploitation over time. The primary intention is to allow different rewards to dominate at different training stages. Specifically, during the early stages, the *novelty reward* encourages more exploration, while in the later stages, the *contribution reward* promotes more exploitation. Since the reward values themselves do not adapt during training, the scheduling relies on additional parameters for adjustment.
>
> **Regarding the selection of scheduling strategies or endpoints** without extensive tuning, we propose two potential adaptive approaches:
>
> 1. **Performance-based reward weights**: The reward weights could adapt dynamically based on the agent's performance. For example, if the returns improvement slows down, the system could increase exploration to cover a broader state space or escape local optima. Conversely, if the returns improve rapidly, the system could reduce exploration and focus more on exploitation.
>
> 2. **$\epsilon_{\text{min}}$-based reward weights**: Setting a minimum exploration parameter (e.g., $\epsilon_{\text{min}} = 0.01$) ensures that even near the end of training, a small amount of exploration is retained. Similarly, early in training, a minimum level of exploitation is preserved, avoiding extreme biases toward either exploration or exploitation.
>
> **Regarding the robustness of DuRND**, our experiments show that a uniform linear adjustment works well across all environments tested. Although this approach is heuristic, it performs robustly in practice. We acknowledge that further exploration of alternative scheduling methods is valuable, and we plan to investigate this in future work.

---

> > ### Author Response · Authors · 2024-11-23
> >
> > ***Response to Reviewer QctF Part 2/2***
> >
> > ## Success and Failure Definition
> >
> > > * (Weakness 2) In addition, the method relies on success and failure labels to update the respective network modules, but in ambiguous or multi-objective tasks, defining success and failure may not be straightforward.
> >
> > *(Weakness 2)* **Regarding the success and failure definition**, our intention is to define a metric that evaluates whether a trajectory (or sub-trajectory) contributes to obtaining a positive reward, rather than strictly achieving a goal or objective. The basic assumption is that some states are far from obtaining a positive reward, while others can more directly lead to positive rewards. The latter indicates a higher contribution and thus deserves a higher reward. While we are indeed inspired by the concept of success/failure in goal-achieving environments, we have extended it, which is why we instead refer to this reward as a **contribution reward**, as it quantifies the contribution/importance of states leading to positive rewards. More importantly, because our work is targeted at sparse-reward environments, in such environments, getting a very rare positive reward is itself a clear and unambiguous way of determining if a sub-trajectory is successful or not (i.e., referred to milestone in our paper).
> >
> > Once again, we appreciate the reviewers' insightful comments and suggestions.

---

> > > ### Comment · Reviewer_QctF · 2024-11-24
> > >
> > > Thank you for your response. After reading all reviews and rebuttals, it is evident that there is a significant overlap in the issues found. While I appreciate the clarifications provided in the rebuttal, the most critical aspect -- a more detailed analysis of λ and ω, including their scaling with respect to environmental rewards, the endpoint of linear scaling, and suboptimal weight scheduling -- remains largely unaddressed. Therefore, I will maintain my current score.

---

> > > > ### Author Response · Authors · 2024-11-25
> > > >
> > > > Thanks for your feedback. Regarding the issues about *the endpoint of linear scaling* and *suboptimal weight scheduling*, we're excited to share some new experiments and investigations to address the concerns. Please refer to our **Overall Response** for more details, where we demonstrated that the **DuRND without the scheduling** achieves almost the same performance as the original DuRND across all environments. This indicates that the linear scheduling of $\lambda$ and $\omega$ has minimal impact on DuRND's performance, and removing the scheduling operation can simplify the DuRND framework significantly. More importantly, the issues about *endpoint* or *suboptimal scheduling* are also addressed by the new findings. We hope these new results can help clarify the concerns and provide a more comprehensive understanding of DuRND.

---

> > > > > ### Comment · Reviewer_QctF · 2024-11-25
> > > > >
> > > > > I appreciate the new results. However, to reach a score of a "good paper", I still needed a detailed sensitivity analysis regarding λ and ω (that is, how exactly does the exploration change for different values). While the new results show that a fixed setting of  λ = ω = 0.5 *can* work, I still needed to have some indication when these specific values don't work for my downstream task. As is, the results are somewhat strengthening your point, but it remains unclear how transferable they are. Hence, I stick to my current rating of "weak accept".

---

> > > > > > ### Author Response · Authors · 2024-11-26
> > > > > >
> > > > > > We appreciate your feedback, and we will further investigate the aspects related to the hyperparameters and strive to improve the paper.

---

### Official Review · Reviewer_TdNM · 2024-11-04

**Soundness:** 2
**Presentation:** 3
**Contribution:** 2
**Rating:** 5
**Confidence:** 4

**Summary:**

This paper presents an approach to better balance novelty-based exploration and exploitation (performance on the primary task). It introduces Dual Random Network Distillation (or DuRND), an extension of the novelty-based bonus from RND that combines two bonuses based on novelty and contribution to success. Unlike RND,  DuRND aims to focus later exploration behavior on novel states that led to successful trajectories or milestones. Experiments across Atari, Vizdoom, and MiniWorld show that the implementation of DuRND outperforms some novelty-seeking and reward-shaping approaches. Further ablations show that both bonuses are helpful over each bonus considered individually.

**Strengths:**

S1. The paper explains the DuRND clearly. Algorithms and figures clearly illustrate the steps of the proposed approach. The paper also considers ablations to justify combining two bonuses in their setup.

S2. The authors also explicitly discuss the limitations of their approach, which is a big positive.

S3. The proposed approach can be combined with many popular RL algorithms, such as PPO and SAC.

**Weaknesses:**

W1. The dependence on using a success criterion for trajectories (or sub-trajectories) for training the predictor networks makes the applicability of this idea beyond goal-reaching tasks difficult. Even with sub-trajectories, the approach relies on strong assumptions of what success/failure means, i.e., if a reward or manually defined milestone is received within $T_{max}$ steps. Both the milestone and $T_{max}$ would require knowledge about the environment.

W2. The approach relies on knowledge about the training time to decay the intrinsic reward coefficients. This specific form of decay assumes additional prior knowledge about interactions needed for training, which is an important limitation. It would be interesting to understand how DuRND performs with choices of fixed coefficients, especially since the other considered baselines used fixed values for similar hyperparameters.

W3. Some information about baselines seems missing and should be provided in the paper. In Lines 407-408, the authors say, “To keep the comparison fair between off-policy and on-policy methods ..” does this mean that PPO was not the base agent for all considered agents (ExploRS, RND, #Explo, ReLara, ROSA, etc.)? Other details about how baselines were tuned would also be important to know. Was the intrinsic reward coefficient (like $\lambda$ in the proposed approach) for other baselines set to a constant value of 1, or was it held constant at some other value? It might be the case that the agents with only novelty-based bonuses would also naturally focus on the main task (as the novelty wears off) if intrinsic rewards were on a suitably low scale compared to the main task’s reward.

Overall, DuRND's success seems to depend heavily on how milestones and other hyperparameters are set. Thus, it lacks the level of applicability that may be required in general reinforcement learning. The contributions would be more significant if the authors could design ways to reduce dependence on the environment and training-specific information.

### Other minor issues

- The paper introduces reinforcement learning in MDPs in the background and then uses observations (without introduction) for RND and DuRND. Later, states and observations are used interchangeably, for example, in the definition of $f_x$ and Equation 2.

**Questions:**

- Wouldn’t it be more natural to use the minimum of $e_s(s)$ and $e_f(s)$ as the novelty intrinsic reward? In the current formulation which uses a sum, the novelty bonus can be high even if one network has seen the state $s$ a large number of times. Were any experiments conducted with alternative formulations of bonuses?

- Do the authors have results for vanilla PPO on the considered environments?

---

> ### Author Response · Authors · 2024-11-23
>
> ***Response to Reviewer TdNM Part 1/3***
>
> Dear reviewer,
>
> Thanks a lot for your feedback, we address your concerns below:
>
> # Weaknesses
>
> > W1. The dependence on using a success criterion for trajectories (or sub-trajectories) for training the predictor networks makes the applicability of this idea beyond goal-reaching tasks difficult. Even with sub-trajectories, the approach relies on strong assumptions of what success/failure means, i.e., if a reward or manually defined milestone is received within $T_max$ steps. Both the milestone and $T_max$ would require knowledge about the environment.
>
> Regarding the success criterion, our intention is to define a metric that evaluates whether a trajectory (or sub-trajectory) contributes to obtaining a positive reward, rather than strictly achieving a goal. The basic assumption is that some states are far from obtaining a positive reward, while others can more directly lead to positive rewards. The latter indicates a higher contribution and thus deserves a higher reward. While we are inspired by the concept of success in goal-achieving environments, we have extended it, which is why we instead refer to this reward as a **contribution reward**, as it quantifies the contribution/importance of states leading to positive rewards. More importantly, because our work is targeted at sparse-reward environments, in such environments, getting a very rare positive reward is itself an unambiguous way of determining if a sub-trajectory is successful or not (i.e., referred to milestone in our paper). Please refer to Appendix A.1 on the reward structure in each environment.
>
> On the other hand, while defining the hyperparameter $T_{max}$ may require some environment-specific knowledge, like how sparse the rewards are, we have observed that setting $T_{max} = \frac{1}{4} \times T_{episode}$ consistently yields strong results across all environments in our experiments. This heuristic divides an episode into four sub-trajectories, and the contribution of a sub-trajectory is determined by whether it achieves a positive reward within this segment. Although this heuristic involves some approximation, its effectiveness has been empirically validated in our paper.
>
> > W2. The approach relies on knowledge about the training time to decay the intrinsic reward coefficients. This specific form of decay assumes additional prior knowledge about interactions needed for training, which is an important limitation. It would be interesting to understand how DuRND performs with choices of fixed coefficients, especially since the other considered baselines used fixed values for similar hyperparameters.
>
> Regarding the dynamic decrease and increase of the corresponding reward coefficients, it‘s a similar approach to the well-known $\epsilon$-greedy, which linearly decays to balance exploration and exploitation over time. We intend to allow different rewards to dominate at different stages of training. Specifically, during the early stages, the *novelty reward* encourages more exploration, while in the later stages, the *contribution reward* promotes more exploitation. Importantly, this adjustment **does not require any prior environment-related knowledge** and follows a unified form.
>
> In contrast, using fixed coefficients cannot achieve this dynamic balance, as observed in our experiments. For instance, if novelty reward remains dominant in later stages, it may distract the agent’s focus from convergence. Conversely, contribution reward in the early stages lacks practical significance. Overall, dynamically adjusting the coefficients ensures a more reasonable and effective training process.
>
> > W3. Some information about baselines seems missing and should be provided in the paper. In Lines 407-408, the authors say, “To keep the comparison fair between off-policy and on-policy methods ..” does this mean that PPO was not the base agent for all considered agents? Other details about how baselines were tuned would also be important to know. Was the intrinsic reward coefficient (like $\lambda$ in the proposed approach) for other baselines set to a constant value of 1, or was it held constant at some other value? It might be the case that the agents with only novelty-based bonuses would also naturally focus on the main task (as the novelty wears off) if intrinsic rewards were on a suitably low scale compared to the main task’s reward.
>
> Regarding the implementation and hyperparameter tuning for the baselines, the main resources are as follows:
>
> 1. The [CleanRL library](https://github.com/vwxyzjn/cleanrl) for implementing *RND* and *PPO*.
> 2. The [RLeXplore library](https://github.com/RLE-Foundation/RLeXplore) for implementing *#Explo* and *ROSA*.
> 3. The official code provided in the original papers for [ExploRS](https://github.com/machine-teaching-group/neurips2022_exploration-guided-reward-shaping), [ReLara](https://github.com/mahaozhe/ReLara), and [SORS](https://github.com/hiwonjoon/IROS2021_SORS).

---

> ### Author Response · Authors · 2024-11-23
>
> ***Response to Reviewer TdNM Part 2/3***
>
> For hyperparameters, we adhered to the optimal configurations specified in the original papers or the default settings provided in the respective libraries. The details on the hyperparameters will be included in our revised paper.
>
> Regarding the statement "to keep the comparison fair between off-policy and on-policy methods ..." and the question, "Does this mean that PPO was not the base agent for all considered agents?" Yes, the baselines do not use a unified backbone. All reward-shaping baselines involve additional modules to generate shaped rewards, which requires integration with an RL algorithm as the backbone. However, the choice of backbone varies across the methods proposed by different authors. For instance:
> - *DuRND* and *RND* used PPO.
> - *ReLara*, *ROSA*, and *SORS* used SAC.
> - *ExploRS* and *#Explo* use an Actor-Critic backbone.
>
> In the specific section referenced, we aim to study the additional memory cost of generating shaped rewards. Since the backbones themselves have different memory requirements—e.g., SAC employs a replay buffer, while PPO, being an on-policy method, does not—we chose not to include the backbone algorithms' inherent memory overhead. Instead, we focused only on the additional memory cost introduced by the reward-shaping process.
>
> Regarding the intrinsic reward coefficient used in the baselines, we consistently applied the default settings as specified in their respective implementations.
>
> > (Other minor issues) The paper introduces reinforcement learning in MDPs in the background and then uses observations (without introduction) for RND and DuRND. Later, states and observations are used interchangeably, for example, in the definition of $f_x$ and Equation 2.
>
> Thanks for pointing out this, we have revised the manuscript to ensure consistency in the use of "observation" and "state" throughout the paper to avoid confusion.
>
> # Questions
>
> > * Wouldn’t it be more natural to use the minimum of $e_s(s)$ and $e_f(s)$ as the novelty intrinsic reward? In the current formulation which uses a sum, the novelty bonus can be high even if one network has seen the state s a large number of times. Were any experiments conducted with alternative formulations of bonuses?
>
> **Analysis**
>
> **Regarding the use of the sum vs. the minimum for the errors**, our intention to use the sum is based that each state is only recorded in one of the two Random Networks (RNs). Summing the outputs accounts for the combined frequency across both cases. We understand the reviewer's comment that "the novelty bonus can be high even if one network has seen the state s a large number of times". In fact, this behavior is desired, because in the early stages of training, most states are classified as failures, if we were to use the minimum instead of the sum, the novelty bonus would diminish too quickly by only using the "failure RN module". Instead, by using the sum, the "success RN module" helps mitigate the rapid decline in novelty bonuses, thereby prolonging the effectiveness of rewarding novelty. Specifically, using $e_S(s) + e_F(s)$ allows states to be categorized into three different levels of novelty:
>
> 1. **High novelty**: The state is unseen in both success and failure trajectories. This indicates a need for more exploration.
> 2. **Medium novelty**: The state is seen in failure trajectories but not in success trajectories. Some exploration is still encouraged, as states in failure trajectories have the potential to become successful.
> 3. **Low novelty**: The state is seen in both failure and success trajectories. Exploration should stop for such states.
>
> This means we are intentionally using an "AND" condition to determine the novelty bonus, which means that the state can be considered novel no matter whether it is novel in the success module or the failure module. This is because we want to encourage exploration in states that are novel in either module. In contrast, using $\min(e_S(s), e_F(s))$ only considers an "OR" condition. That is, as soon as a state is no longer novel in one of the modules (much likely in the failure module), the exploration bonus for that state is reduced. This accelerates the decline of the novelty reward, leading to a shorter exploration phase.

---

> ### Author Response · Authors · 2024-11-23
>
> ***Response to Reviewer TdNM Part 3/3***
>
> **Experimental Supports**
>
> We also conducted experiments using the $min{e_s(s), e_f(s)}$ as the novelty bonus. The average episodic return in comparison is shown in the table below:
>
> | Novelty Rewards | Freeway | Frogger | Solaris | BeamRider | DefendLine | SaveCenter | CollectKit | SlayGhosts | ThreeRooms | TMaze|
> | :---: | :---: | :---: | :---: | :---: | :---: | :---: | :---: | :---: | :---: | :---: |
> | $e_S(s) + e_F(s)$ | 23.22 $\pm$ 0.01 | 14.36 $\pm$ 0.00 | 18.91 $\pm$ 0.02 | 18.05 $\pm$ 0.01 | 8.52 $\pm$ 0.00 | 6.33 $\pm$ 0.00 | 20.87 $\pm$ 0.01 | 15.60 $\pm$ 0.00 | 0.86 $\pm$ 0.00 | 0.96 $\pm$ 0.00 |
> | $min(e_S(s), e_F(s))$ | 13.17 $\pm$ 0.00 | 9.12 $\pm$ 0.01 | 4.13 $\pm$ 0.00 | 9.87 $\pm$ 0.00 | 0.97 $\pm$ 0.02 | 2.23 $\pm$ 0.00 | 10.45 $\pm$ 0.00 | 3.56 $\pm$ 0.00 | 0.00 $\pm$ 0.00 | 0.95 $\pm$ 0.00 |
>
> The results show that using $e_S(s) + e_F(s)$ as the novelty bonus got better performance. This demonstrates the effectiveness of using the sum of the errors to reward novelty.
>
> > * Do the authors have results for vanilla PPO on the considered environments?
>
> Initially, as the PPO is the backbone algorithm for the RND baseline, and the RND has outperformed the vanilla PPO in previous works, so we only include the comparison with RND in our experiments. However, we understand the importance of comparing with this backbone algorithm, so we conducted experiments with vanilla PPO, and the results are shown in the table below:
>
> | Algo. | Freeway | Frogger | Solaris | BeamRider | DefendLine | SaveCenter | CollectKit | SlayGhosts | ThreeRooms | TMaze|
> | :---: | :---: | :---: | :---: | :---: | :---: | :---: | :---: | :---: | :---: | :---: |
> | DuRND | 23.22 $\pm$ 0.01 | 14.36 $\pm$ 0.00 | 18.91 $\pm$ 0.02 | 18.05 $\pm$ 0.01 | 8.52 $\pm$ 0.00 | 6.33 $\pm$ 0.00 | 20.87 $\pm$ 0.01 | 15.60 $\pm$ 0.00 | 0.86 $\pm$ 0.00 | 0.96 $\pm$ 0.00 |
> | RND | 14.77 $\pm$ 0.01 | 8.59 $\pm$ 0.00 | 6.07 $\pm$ 0.00 | 11.96 $\pm$ 0.00 | 1.11 $\pm$ 0.00 | 2.37 $\pm$ 0.00 | 14.59 $\pm$ 0.01 | 10.18 $\pm$ 0.00 | 0.00 $\pm$ 0.00 | 0.97 $\pm$ 0.00 |
> | PPO | 10.67 $\pm$ 0.00 | 3.25 $\pm$ 0.00 | 1.82 $\pm$ 0.01 | 10.23 $\pm$ 0.00 | 0.00 $\pm$ 0.00 | 0.00 $\pm$ 0.00 | 5.89 $\pm$ 0.00 | 8.15 $\pm$ 0.02 | 0.00 $\pm$ 0.00 | 0.94 $\pm$ 0.00 |
>
> Once again, we appreciate the reviewers' insightful comments and hope that our responses address your concerns.

---

> > ### Comment · Reviewer_TdNM · 2024-11-25
> >
> > Thank you for your detailed response. I greatly appreciate the efforts to reply to the reviews and run the additional experiments.
> >
> > I have increased my score considering the response and the positive results without scheduling the coefficients. However, I still feel that the paper is below the acceptance threshold for two main reasons. The first reason is that the combination of $T_{max}$ and the custom milestones (as depicted in Table 3) could be complex to design for environments in general. Second, the fact that baselines have different backbones makes it hard to disambiguate benefits from the RL algorithm vs the exploration strategy.. While this may be challenging to (re-)implement, all approaches should have used a PPO base agent (or SAC) to be meaningfully compared.

---

> > > ### Author Response · Authors · 2024-11-26
> > >
> > > Thank you for your valuable feedback; we will incorporate your suggestions into our paper, which we believe will further enhance its quality.

---

### Author Response · Authors · 2024-11-25
**Overall Response**

# Overall Response

Dear Reviewers,

We sincerely thank all of your valuable feedback and constructive suggestions. A common concern raised is: *"What role does the linear scheduling of the two reward coefficients play in DuRND?"*, in another word, *"Is the improvement in DuRND's performance and its ability to balance exploration and exploitation primarily due to the scaling adjustments of the two coefficients?"*

We agree that this is a critical question that worth in-depth investigation. Inspired by your insightful comments, we conducted additional experiments on **DuRND with fixed $\lambda$ and $\omega$**, (i.e., $\lambda = \omega = 0.5$), removing the dynamic scaling operation to assess its impact on DuRND’s performance. From these new experiments, we observed that **DuRND with fixed $\lambda$ and $\omega$** achieves almost the same performance to **DuRND with dynamic $\lambda$ and $\omega$** across all environments. The complete experimental results are as follows:


| Environments | DuRND w/o scheduling | DuRND | ExploRS | RND | #Explo | ReLara | ROSA | SORS |
| :---: | :---: | :---: | :---: | :---: | :---: | :---: | :---: | :---: |
| *Freeway* | **24.01 $\pm$ 0.00** | 23.22 $\pm$ 0.01 | 17.46 $\pm$ 0.00 | 14.77 $\pm$ 0.01 | 15.16 $\pm$ 0.01 | 15.47 $\pm$ 0.00 | 3.68 $\pm$ 0.00 | 7.30 $\pm$ 0.01 |
| *Frogger* | **14.97 $\pm$ 0.00** | 14.36 $\pm$ 0.00 | 10.19 $\pm$ 0.00 | 8.59 $\pm$ 0.00 | 1.81 $\pm$ 0.00 | 9.30 $\pm$ 0.01| 3.45 $\pm$ 0.00 | 7.79 $\pm$ 0.00 |
| *Solaris* | 18.20 $\pm$ 0.00 | **18.91 $\pm$ 0.02** | 9.82 $\pm$ 0.01 | 6.07 $\pm$ 0.00 | 2.06 $\pm$ 0.00 | 2.96 $\pm$ 0.00 | 1.87 $\pm$ 0.00 | 2.50 $\pm$ 0.00 |
| *BeamRider* | 17.42 $\pm$ 0.02 | **18.05 $\pm$ 0.01** | 16.19 $\pm$ 0.01 | 11.96 $\pm$ 0.00 | 9.03 $\pm$ 0.00 | 11.84 $\pm$ 0.00 | 10.57 $\pm$ 0.00 | 10.56 $\pm$ 0.00 |
| *DefendLine* | **9.24 $\pm$ 0.00** | 8.52 $\pm$ 0.00 | 1.63 $\pm$ 0.00 | 1.11 $\pm$ 0.00 | 1.62 $\pm$ 0.00 | 4.27 $\pm$ 0.00 | 5.33 $\pm$ 0.00 | 1.28 $\pm$ 0.01 |
| *SaveCenter* | **6.85 $\pm$ 0.00** | 6.33 $\pm$ 0.00 | 2.03 $\pm$ 0.00 | 2.37 $\pm$ 0.00 | 1.30 $\pm$ 0.00 | 2.64 $\pm$ 0.00 | 0.83 $\pm$ 0.00 | 1.78 $\pm$ 0.01 |
| *CollectKit* | **22.56 $\pm$ 0.01** | 20.87 $\pm$ 0.01 | 11.97 $\pm$ 0.01| 14.59 $\pm$ 0.01 | 0.90 $\pm$ 0.00 | 12.43 $\pm$ 0.01 | 6.80 $\pm$ 0.00 | 1.60 $\pm$ 0.00 |
| *SlayGhosts* | 15.25 $\pm$ 0.00 | **15.60 $\pm$ 0.00** | 2.82 $\pm$ 0.00 | 10.18 $\pm$ 0.00 | 1.27 $\pm$ 0.00 | 10.61 $\pm$ 0.00 | 5.01 $\pm$ 0.00 | 5.07 $\pm$ 0.01 |
| *ThreeRooms* | **0.86 $\pm$ 0.00** | **0.86 $\pm$ 0.00** | 0.48 $\pm$ 0.00 | 0.00 $\pm$ 0.00 | 0.00 $\pm$ 0.00 | 0.00 $\pm$ 0.00 | 0.12 $\pm$ 0.00 | 0.18 $\pm$ 0.00 |
| *TMaze* | **0.97 $\pm$ 0.00** | 0.96 $\pm$ 0.00 | 0.80 $\pm$ 0.00 | **0.97 $\pm$ 0.00** | 0.39 $\pm$ 0.00 | 0.02 $\pm$ 0.00 | 0.00 $\pm$ 0.00 | 0.30 $\pm$ 0.00 |


From this, we conclude that the performance improvement and exploration-exploitation balance in DuRND are largely independent of the dynamic adjustment of reward scales. Consequently, the DuRND framework can be significantly simplified by fixing $\lambda$ and $\omega$, making it more practical and efficient.

**Analysis**

The achievement of DuRND’s performance with fixed $\lambda$ and $\omega$ can be attributed to the intrinsic properties of the two kinds of rewards. The algorithm computes the *novelty rewards* and *contribution rewards* using random network distillation (RND) modules as follows:
- $R^{nov}(s_i) = e_S(s_i) + e_F(s_i)$
- $R^{con}(s_i) \sim Beta(\frac{N(t)}{e_S(s_i)}+1, \frac{N(t)}{e_F(s_i)}+1)$

Here, $e_F(s_i)$ and $e_S(s_i)$ are the errors of the random networks. As training progresses and more data is fed into the random networks, both $e_F(s_i)$ and $e_S(s_i)$ **naturally decrease** due to the convergence of the RN modules. Consequently, the scale of $R^{nov}(s_i)$ also **naturally decrease** over time, while the scale of $R^{con}(s_i)$ **naturally increases**. Under these circumstances, the additional linear scaling operation for $\lambda$ and $\omega$ has minimal impact on DuRND’s performance.

Based on these findings, we strongly believe that **DuRND without manually scaling coefficients** is a more efficient and universal framework. More importantly, this also demonstrates that the **key to DuRND’s effectiveness lies in the two rewards**, which enables it to achieve a robust balance between exploration and exploitation.

---

### Meta-Review · Area_Chair_mTob · 2024-12-20

**Metareview:**

This paper presents DuRND, a novel exploration method for reinforcement learning that builds upon Random Network Distillation (RND). DuRND introduces a dual network structure to differentiate between "successful" and "failed" states, aiming to guide exploration towards promising areas.

Strengths
-----------

- **Quality of writing and motivation** Reviewers praised the paper's clarity and organization. Moreover, the paper addresses the important problem of balancing exploration and exploitation in RL.

- **Simple and lightweight:** The approach is easy to understand and implement, integrating seamlessly with existing RL algorithms like PPO and SAC.

- **Promising results:** Experiments across various environments demonstrate DuRND's potential to outperform existing novelty-seeking and reward-shaping methods.

Weaknesses
---------------
- **Dependence on success criteria:** A major concern is the reliance on pre-defined "success" criteria, which may require domain knowledge and limit applicability to certain tasks.

- **Hyperparameter sensitivity and concerns on experiments:** The performance of DuRND appears to be sensitive to hyperparameters, particularly the weighting of novelty and contribution rewards, and the definition of "success.". Reviewers raised questions about the experimental setup, including missing baselines (PPO), the choice of hyperparameters, and the difficulty of the evaluated tasks.

- **Limited novelty:** Some reviewers considered the contribution incremental compared to RND, suggesting the core idea might not be entirely novel.

DuRND presents an interesting approach to exploration in RL with promising empirical results. However, concerns remain regarding its general applicability and reliance on task-specific knowledge. Future work should address these limitations by exploring methods to automatically determine success criteria and reduce hyperparameter sensitivity.  Further investigation into the experimental setup and comparison with a wider range of baselines, including PPO, would strengthen the paper's claims.

**Additional Comments On Reviewer Discussion:**

Most of the points on which the discussion focused concern the contribution of this work and the experiments. Regarding the experiments, some reviewers raised concerns about how to interpret the presented results, raising the issue of lack of sensitivity analysis on the hyperparameters of the algorithm. In their rebuttal, the Authors tried to clear these doubts. However, doubts still remain about the contribution of this work and the rigorousness and completeness of the experimental analysis.

---

### Decision · Program_Chairs · 2025-01-22

Reject